# Approximation-Generalization Trade-offs under (Approximate) Group Equivariance

**Mircea Petrache**[*]           **Shubhendu Trivedi**[†]

## Abstract

The explicit incorporation of task-specific inductive biases through symmetry has emerged as a general design precept in the development of high-performance machine learning models. For example, group equivariant neural networks have demonstrated impressive performance across various domains and applications such as protein and drug design. A prevalent intuition about such models is that the integration of relevant symmetry results in enhanced generalization. Moreover, it is posited that when the data and/or the model may only exhibit *approximate* or *partial* symmetry, the optimal or best-performing model is one where the model symmetry aligns with the data symmetry. In this paper, we conduct a formal unified investigation of these intuitions. To begin, we present general quantitative bounds that demonstrate how models capturing task-specific symmetries lead to improved generalization. In fact, our results do not require the transformations to be finite or even form a group and can work with partial or approximate equivariance. Utilizing this quantification, we examine the more general question of model mis-specification i.e. when the model symmetries don't align with the data symmetries. We establish, for a given symmetry group, a quantitative comparison between the approximate/partial equivariance of the model and that of the data distribution, precisely connecting model equivariance error and data equivariance error. Our result delineates conditions under which the model equivariance error is optimal, thereby yielding the best-performing model for the given task and data. Our results are the most general results of their type in the literature.

## 1   Introduction

It is a common intuition that machine learning models that explicitly incorporate task-specific symmetries tend to exhibit superior generalization. The rationale is simple: if aspects of the task remain unchanged or change predictably when subject to certain transformations, then, a model without knowledge of them would require additional training samples to learn to disregard them. While the general idea is by no means new in machine learning, it has experienced a revival in interest due to the remarkable successes of group equivariant convolutional neural networks (GCNNs) [14, 38, 29, 16, 13, 60, 8, 34, 24, 66]. Such networks hard-code equivariance to the action of an algebraic group on the inputs and have shown promise in a variety of domains [55, 5, 42, 2, 28, 62, 25, 52, 18, 61, 68], in particular in those where data is scarce, but exhibits clear symmetries. Further, the benefits of encoding symmetries in this manner extend beyond merely enhancing generalization. Indeed, in

---

[*]UC Chile, Fac. de Matemáticas, & Inst. de Ingeniería Matematica y Computacional, Av. Vicuña Mackenna 4860, Santiago, 6904441, Chile. `mpetrache@mat.uc.cl`.

[†]`shubhendu@csail.mit.edu`.

37th Conference on Neural Information Processing Systems (NeurIPS 2023).

numerous applications in the physical sciences, neglecting to do so could lead to models producing unphysical outputs. However, the primary focus of most of the work in the area has been on estimating functions that are *strictly* equivariant under a group action. It has often been noted that this may impose an overly stringent constraint as real-world data seldom exhibits perfect symmetry. The focus on strict equivariance is partly due to convenience: the well-developed machinery of group representation theory and non-commutative harmonic analysis can easily be marshalled to design models that are exactly equivariant. A general prescriptive theory for such models also exists [29, 16]. In a certain sense, the basic architectural considerations of such models are fully characterized.

Recent research has begun to explore models that enforce only partial or approximate equivariance [40, 59, 23, 56]. Some of these works suggest interpolating between models that are *exactly* equivariant and those that are *fully flexible*, depending on the task and data. The motivation here is analogous to what we stated at the onset. If the data has a certain symmetry – whether exact, approximate or partial – and the model's symmetry does not match with it, its performance will suffer. We would expect that a model will perform best when its symmetry is correctly specified, that is, it aligns with the data symmetry.

Despite increased research interest in the area, the theoretical understanding of these intuitions remains unsatisfactory. Some recent work has started to study improved generalization for exactly equivariant/invariant models (see section 2). However, for the more general case when the data and/or model symmetries are only approximate or partial, a general treatment is completely missing. In this paper, we take a step towards addressing this gap. First, we show that a model enforcing task-specific invariance/equivariance exhibits better generalization. Our theoretical results subsume many existing results of a similar nature. Then we consider the question of model mis-specification under partial/approximate symmetries of the model and the data and prove a general result that formalizes and validates the intuition we articulated above.

We summarize below the main contributions of the paper:

- We present quantitative bounds, the most general to date, showing that machine learning models enforcing task-pertinent symmetries in an equivariant manner afford better generalization. In fact, our results do not require the set of transformations to be finite or even be a group. While our investigation was initially motivated by GCNNs, our results are not specific to them.

- Using the above quantification, we examine the more general setting when the data and/or model symmetries are *approximate* or *partial* and the model might be *misspecified* relative to the data symmetry. We rigorously formulate the relationship between model equivariance error and data equivariance error, teasing out the precise conditions when the model equivariance error is optimal, that is, provides the best generalization for given data. To the best of our knowledge, this is the most general result of its type in the literature.

## 2   Related Work

As noted earlier, the use of symmetry to encode inductive biases is not a new idea in machine learning and is not particular to neural networks. One of the earliest explorations of this idea in the context of neural networks appears to be [46]. Interestingly, [46] originated in response to the group invariance theorems in Minsky & Papert's *Perceptrons* [36], and was the first paper in what later became a research program carried out by Shawe-Taylor & Wood, building more general symmetries into neural networks [63, 48, 49, 65, 64], which also included PAC style analysis [50]. While Shawe-Taylor & Wood came from a different perspective and already covered discrete groups [14, 38], modern GCNNs go beyond their work [29, 16, 13, 60, 8, 34, 24]. Besides, GCNNs [14] were originally proposed as generalizing the idea of the classical convolutional network [32], and seem to have been inspired by symmetry considerations from physics [15]. Outside neural networks, invariant/equivariant methods have also been proposed in the context of support vector machines [43, 57], general kernel methods [39, 26], and polynomial-based feature spaces [44].

On the theoretical side, the assertion that invariances enhance generalization can be found in many works, going back to [47]. It was argued by [1] that restricting a classifier to be invariant can not increase its VC dimension. Other examples include [3, 4, 51, 41]. The argument in these, particularly [51, 41] first characterizes the input space and then proceeds to demonstrate that certain data transformations shrink the input space for invariant models, simplifying the input and improving generalization. Some of our results generalize their results to the equivariant case and for more general transformations.

Taking a somewhat different tack, [22] showed a strict generalization benefit for equivariant linear models, showing that the generalization gap between a least squares model and its equivariant counterpart depends on the dimension of the space of anti-symmetric linear maps. This result was adapted to the kernel setting in [19]. [9] studied sample complexity benefits of invariance in a kernel ridge regression setting under a geometric stability assumption, and briefly discussed going beyond group equivariance. [53] expands upon [9] by characterizing minimax optimal rates for the convergence of population risk in a similar setting when the data resides on certain manifolds. [35] characterizes the benefits of invariance in overparameterized linear models, working with invariant random features projected onto high-dimensional polynomials. Benefits of related notions like data augmentation and invariant averaging are formally shown in [33, 12]. Using a standard method from the PAC-Bayesian literature [37] and working on the intertwiner space, [7] provide a PAC-Bayesian bound for equivariant neural networks. Some improved generalization bounds for transformation-invariant functions are proposed in [67], using the notion of an induced covering number. A work somewhat closer to some of our contributions in this paper is [20], which gives PAC bounds for (exact) group equivariant models. However, we note that the proof of the main theorem in [20] has an error (see the Appendix for a discussion), although the error seems to be corrected in the author's dissertation [21]. An analysis that somewhat overlaps [20], but goes beyond simply providing worst-case bounds was carried out by [45].

Finally, recent empirical work on approximate and partial equivariance includes [40, 59, 23, 56]. These make the case that enforcing strict equivariance, if misspecified, can harm performance, arguing for more flexible models that can learn the suitable degree of equivariance from data. However, there is no theoretical work that formalizes the underlying intuition.

## 3 Preliminaries

In this section, we introduce basic notation and formalize some key notions that will be crucial to ground the rest of our exposition.

### 3.1 Learning with equivariant models

**Learning problem.** To begin, we first describe the general learning problem which we adapt to the equivariant case. Let's say we have an input space $\mathcal{X}$ and an output space $\mathcal{Y}$, and assume that the pairs $Z = (X, Y) \in \mathcal{X} \times \mathcal{Y}$ are random variables having distribution $\mathcal{D}$. Suppose we observe a sequence of $n$ i.i.d pairs $Z_i = (X_i, Y_i) \sim \mathcal{D}$, and want to learn a function $\tilde{f} : \mathcal{X} \to \mathcal{Y}$ that predicts $Y$ given some $X$. We will consider that $\tilde{f}$ belongs to a class of functions $\widetilde{\mathcal{F}} \subset \{\tilde{f} : \mathcal{X} \to \mathcal{Y}\}$ and that we work with a loss function $\ell : \mathcal{X} \times \mathcal{Y} \to [0, \infty)$. To fully specify the learning problem we also need a *loss class*, comprising of the functions $f(x, y) := \ell(\tilde{f}(x), y) : \mathcal{X} \times \mathcal{Y} \to [0, \infty)$. Using these notions, we can define:

$$\mathcal{F} := \{f(x, y) := \ell(\widetilde{f}(x), y) : \tilde{f} \in \widetilde{\mathcal{F}}\}, \quad Pf := \mathbb{E}[f(X, Y)], \quad P_n f := \frac{1}{n} \sum_{i=1}^{n} f(X_i, Y_i).$$

The quantities $Pf$ and $P_n f$ are known as the risk and the empirical risk associated with $\widetilde{f}$, respectively. In practice we have access to $P_n f$, and we estimate $Pf$ by it. One of the chief quantities of interest in such a setup is the worst-case error of this approximation on a sample $\{Z_i\} = \{Z_i\}_{i=1}^{n}$, that is, the generalization error:

$$\mathsf{GenErr}(\mathcal{F}, \{Z_i\}, \mathcal{D}) := \sup_{f \in \mathcal{F}} (Pf - P_n f).$$

**Group actions.** We are interested in the setting where a group $G$ acts by measurable bijective transformations over $\mathcal{X}$ and $\mathcal{Y}$, which can also be seen as a $G$-action over $Z$, denoted

by $g \cdot Z = (g \cdot X, g \cdot Y)$, where $g \in G$. Such actions can also be applied to points $(x, \widetilde{f}(x))$ in the graph of functions $\widetilde{f}$, transforming any $\tilde{f} \in \widetilde{\mathcal{F}}$ into $g \cdot \tilde{f} : x \mapsto g^{-1} \cdot \tilde{f}(g \cdot x)$. The set $G \cdot \tilde{f} := \{g \cdot \tilde{f} : g \in G\}$ forms the orbit of $\tilde{f}$ under the action of $G$. Note that although we treat $G$ as an abstract group throughout, we will sometimes abuse notation and use the same letter to also refer to the representations of $G$ as transformations over the spaces in which $X, Y, Z, \widetilde{f}, f$ live.

**Equivariant models.** We are now in a position to specify the learning setup with equivariant models. We say that $\widetilde{f} \in \widetilde{\mathcal{F}}$ is equivariant, if $G \cdot \widetilde{f} = \{\widehat{f}\}$. Further, we say that the hypothesis class is $G$-equivariant if all $\widetilde{f} \in \widetilde{\mathcal{F}}$ are equivariant. Note that if the $G$-action over $Y$ is trivial, that is, if we have $g \cdot Y = Y$ for all $g \in G, y \in Y$), then $\widetilde{f}$ being equivariant reduces to requiring $\widetilde{f}(g \cdot x) = \widetilde{f}(x)$ for all $g, x$. In such a case the model $\tilde{f}$ is said to be **invariant**. Also note that $\tilde{f}(x)$ being equivariant corresponds to $f(x,y) = \ell(\widehat{f}(x), y)$ being invariant under some suitable product $G$-action over $\mathcal{X} \times \mathcal{Y}$.

**Averaging over a set of transformations.** Let $G$ be a set of transformations of $\mathcal{Z}$. The $G$-averaged generalization error for arbitrary $\mathcal{F}$ by taking averages $\mathbb{E}_g$ with respect to the natural probability measure $\mathbb{P}_G$[3] on $G$ is as follows:

$$\mathsf{GenErr}^{(G)}(\mathcal{F}, \{Z_i\}, \mathcal{D}) := \sup_{f \in \mathcal{F}} (\mathbb{E}_g \mathbb{E}_Z[f(g \cdot Z)] - \frac{1}{n} \sum_{i=1}^{n} \mathbb{E}_g f(g \cdot Z_i)),$$

Note that the above equals $\mathsf{GenErr}(\mathcal{F}, \{Z_i\}, \mathcal{D})$ if $\mathcal{F}$ is composed of invariant functions.

**Data augmentation/Orbit averaging.** Following our discussion above, it might be instructive to consider the case of data augmentation. Note that a data distribution which is invariant under some $G$-action can be created from any $\mathcal{D}$ by averaging over $G$-orbits: we take $g$ to be a random variable uniformly distributed over a compact set of transformations $G$. Then let $\mathcal{D}^{(G)} := \frac{1}{|G|} \int_G g \cdot \mathcal{D}$. This is the distribution of the dataset obtained from $\mathcal{D}$ under $G$-augmentation. Then we have by change of variables:

$$\mathsf{GenErr}^{(G)}(\mathcal{F}, \{Z_i\}, \mathcal{D}) = \mathsf{GenErr}(\mathcal{F}, \{\mathsf{E}_g[g \cdot Z_i]\}, \mathcal{D}^{(G)}). \tag{3.1}$$

Note that the notions of orbit and averaging used above do not rely on the property that $G$ is a group in order to be well-defined.

## 3.2 Partial and Approximate Equivariance

Since our analysis treats the general case where the model and/or data symmetries are not strict, we now state some basic attendant notions.

**Error to equivariance function.** We first need a measure to quantify how far off we are from perfect equivariance. To do so, we can define a function in terms of $f(x, y)$ as:

$$\mathsf{ee} : \mathcal{F} \times G \to [0, +\infty), \quad \mathsf{ee}(f, g) = \|g \cdot f - f\|_\infty.$$

The crucial property here is that $\mathsf{ee}(f, g) = 0$ iff $g \cdot f = f$. Note that one could use other distances or discrepancies (e.g. like the $\ell_2$ norm) directly in terms of $\widetilde{f}$, using which we can measure the error between $g \cdot \tilde{f}$ and $\tilde{f}$.

**More general sets of transformations.** We can use $\mathsf{ee}(\cdot)$ to precisely specify the notions of partial and approximate equivariance that we work with. We consider the general case where the underlying transformations on $(X, Y)$ still have a common label set $G$ and are assumed to be bijective over $\mathcal{X}$ and $\mathcal{Y}$, but don't necessarily form a group. For $\epsilon > 0$ and $f \in \mathcal{F}$ we consider the $\epsilon$-stabilizer

$$\mathsf{Stab}_\epsilon(f) := \{g \in G : \mathsf{ee}(f, g) \le \epsilon\}.$$

It is fairly easy to see that this subset is automatically a group for $\epsilon = 0$, but most often will not be when $\epsilon > 0$. The latter corresponds to *partial symmetry* and thus to the setting

---

[3]We take $\mathbb{P}_G$ to be the uniform measure over $G$ if $G$ is finite, and the normalization of the Haar measure of $\overline{G}$ to $G$ if $G \subset \overline{G}$ is a positive measure subset of an ambient locally compact group $\overline{G}$.

of **partial equivariance**. On the other hand, if we consider specific hypotheses such that for some $\epsilon > 0$ we have $\mathsf{Stab}_\epsilon(f) = G$ for all $f \in \mathcal{F}$, then we are working in the setting of **approximate equivariance** [59, 56].

**Conditions on $G$.** We conclude this section with a remark on the nature of $G$ we consider in our paper. In our bounds, we first restrict to **finite** $G$. However, this setting can be extended to **compact** groups which are doubling with respect to a metric that respects the composition operation. That is, a distance $\mathsf{d}_G$ over $G$, such that $\mathsf{d}_G(gh, g'h) = d(g, g')$. The main examples we have in mind are compact Lie groups such as $S^1, SO(n), O(n)$, and their finite subgroups like $\mathbb{Z}/N\mathbb{Z}$, as well as their quotients and products.

## 4 PAC bounds for $G$-equivariant hypotheses

In this section, we study the generalization properties of equivariant models. More specifically, we derive PAC bounds for both exact and approximately G-equivariant hypotheses. But first, we present some preliminaries related to PAC bounds in what follows.

**PAC learning bounds.**

We follow the standard approach (such as that outlined in [10]). We start with the following concentration bound, where $\mathcal{F}$ is composed of functions with range $[-M/2, M/2]$:

$$\mathbb{P}\left[\sup_{\mathcal{F}} |(P - P_n)f| \geq \mathcal{R}(\mathcal{F}_Z) + \epsilon\right] \leq 2\exp\left(-\frac{\epsilon^2 n}{M}\right), \tag{4.1}$$

where $\mathcal{R}_n(\mathcal{F})$ denotes the Rademacher complexity, which is given by

$$\mathcal{R}(\mathcal{F}_Z) := \mathbb{E}_\sigma \sup_{\mathcal{F}} \frac{1}{n} \sum_{i=1}^{n} \sigma_i f(Z_i),$$

and where $Z_1, \ldots, Z_n \sim \mathcal{D}$ are $n$ i.i.d. samples and $\sigma = (\sigma_1, \ldots, \sigma_n)$ denotes the so-called Rademacher variable, which is a uniformly distributed random variable over $\{-1, 1\}^n$, independent of the $Z_i$'s. $\mathcal{R}(\mathcal{F}_Z)$ can be bounded using a classical technique for bounding the expected suprema of random processes indexed by the elements of a metric space, variously called the Dudley entropy integral or the chaining bound. The result below was proven in [45, Lemma 3] (also see slightly stronger results in [58, Thm. 17] or [6, Thm. 1.1])

$$\mathcal{R}(\mathcal{F}_Z) \leq 4 \inf_{\alpha > 0} \left(\alpha + \frac{3}{\sqrt{n}} \int_\alpha^{\mathsf{diam}(\mathcal{F})} \sqrt{\ln \mathcal{N}(\mathcal{F}, t, \|\cdot\|_\infty)} dt\right). \tag{4.2}$$

Recall that for a metric space $(X, \mathsf{d}_X)$, the **covering number** $\mathcal{N}(X, \epsilon, \mathsf{d}_X)$ is the smallest number of balls of radius $\epsilon$ required to cover $X$. In (4.2) we use the cover numbers of $\mathcal{F}$ in supremum distance, i.e. the distance between two functions $f, g \in \mathcal{F}$, $\|f - g\|_\infty := \sup_{z \in \mathcal{Z}} |f(z) - g(z)|)$. It is known that by rescaling the fundamental estimate due to Kolmogorov-Tikhomirov [54, eq. 238, with $s = 1$, and eq. 1], and under the mild assumption that $\mathcal{F}$ is composed of 1-Lipschitz[4] functions on $\mathcal{Z}$ with values in an interval $[-M/2, M/2]$, for a centralizable[5] metric space $\mathcal{Z}$, the following holds

$$\mathcal{N}(\mathcal{Z}, 2\epsilon) \leq \log_2 \mathcal{N}(\mathcal{F}, \epsilon, \|\cdot\|_\infty) \leq \log_2\left(\frac{M}{\epsilon} + 1\right) + \mathcal{N}(\mathcal{Z}, \epsilon/2). \tag{4.3}$$

Before we start stating our results, we need to define a few more notions:

**Doubling and Hausdorff dimensions.** If in a metric space $(X, \mathsf{d}_X)$, every ball of radius $R$ can be covered by at most $\lambda$ balls of radius $R/2$, the space has doubling dimension $\mathsf{ddim}(X) = \log_2 \lambda$. The doubling dimension coincides with the usual notion of (Hausdorff) dimension $\mathsf{dim}X$, i.e. $\mathsf{ddim}X = \mathsf{dim}X$, in case $X$ is a compact manifold with bounded injectivity radius, in particular it equals $d$ if $X \subset \mathbb{R}^D$ is a regular $d$-dimensional submanifold or if $D = d$ and $X$ is a regular open subset such as a ball or a cube.

---

[4]All of our formulas generalize to classes of $\ell$-Lipschitz functions for general $\ell > 0$: the change amounts to applying suitable rescalings, see the Appendix for more details.

[5]This mild condition signifies that for any open set $U$ of diameter at most $2r$ there exists a point $x^0$ so that $U$ is contained in $B(x^0, r)$.

**Discrete metric spaces.** A metric space $(X, \mathsf{d}_X)$ is **discrete** if it has strictly positive **minimum separation distance** $\delta_X := \min_{x \neq x' \in X} \mathsf{d}_X(x, x') > 0$. We then have

$$\mathcal{N}(X, \epsilon) = \mathcal{N}(X, \min\{\epsilon, \delta_X\}), \tag{4.4}$$

which is especially relevant for **finite groups** $G$ endowed with the **word distance** (i.e. $\mathsf{d}_G(g, g')$, which is the minimum number of generators required to express $g^{-1}g'$), for which $\delta_G = 1$. Now, note that as a straightforward consequence of the definition, there exists a universal $c > 0$ such that[30, 31]:

$$\mathcal{N}(\mathcal{Z}, \epsilon) \leq \left(\frac{2\mathsf{diam}(\mathcal{Z})}{\epsilon}\right)^{\mathsf{ddim}(\mathcal{Z})}. \tag{4.5}$$

and by (4.3) we get the simplified bound (where implicit constants are universal):

$$\ln \mathcal{N}(\mathcal{F}, \epsilon, \|\cdot\|_\infty) \lesssim \left(\frac{4\mathsf{diam}(\mathcal{Z})}{\epsilon}\right)^{\mathsf{ddim}(\mathcal{Z})}. \tag{4.6}$$

By all the above considerations, we can bound the generalization error as follows:

**Proposition 4.1.** *Assume that $d = \mathsf{ddim}(\mathcal{Z}) > 2$, and $0 < \delta < 1/2$, and let $D := \mathsf{diam}(\mathcal{Z})$. Then for any probability distribution $\mathcal{D}$ of data over $\mathcal{Z}$, with notation as in the beginning of Section 4, the following holds with probability at least $1 - \delta$:*

$$\mathsf{GenErr}(\mathcal{F}, \{Z_i\}, \mathcal{D}) \lesssim \frac{4^d d}{d-2}\left(\frac{D^d}{n}\right)^{1/d} + n^{-1/2}\sqrt{\|\mathcal{F}\|_\infty \log(2/\delta)},$$

*the implicit constant is independent of $\delta, n$; only depending on $\mathcal{Z}$ through the constants in* (4.5).

Due to space constraints, all our proofs are relegated to the Appendix. With the above background, we now first show generalization bounds under the more general notions of equivariance we have discussed.

## 4.1 Generalization bounds improvement under partial or approximate equivariance

In this section, we prove generalization error bounds with the notations and definitions from Section 3.2. We consider the following sets of transformations:

$$\mathsf{Stab}_\epsilon = \mathsf{Stab}_\epsilon(\mathcal{F}) := \bigcap_{f \in \mathcal{F}} \mathsf{Stab}_\epsilon(f).$$

We note that the strict equivariance case is recovered if, for $\epsilon = 0$, we have $\mathsf{Stab}_0 = G$.

**Proposition 4.2.** *Let $\mathsf{Stab}_\epsilon$ be as above, and assume that $|\mathsf{Stab}_\epsilon| > 0$. let $\mathcal{Z}_\epsilon^0 \subset \mathcal{Z}$ be a measurable choice of $\mathsf{Stab}_\epsilon$-orbit representatives for points in $\mathcal{Z}$, and let $\iota_\epsilon^0 : \mathcal{Z} \to \mathcal{Z}_\epsilon^0$ be the measurable map that associates to each $z \in \mathcal{Z}$ its representative in $\mathcal{Z}_\epsilon^0$. Let $\mathcal{F}_\epsilon^0 := \{f|_{\mathcal{Z}_\epsilon^0} : f \in \mathcal{F}\}$ and let $\iota_\epsilon^0(\mathcal{D})$ represent the image measure of $\mathcal{D}$. Then for each $n \in \mathbb{N}$, if $\{Z_i\}_{i=1}^n$ are i.i.d. samples with $Z_i \sim \mathcal{D}$ and $Z_{i,\epsilon}^0 := \iota_\epsilon^0 \circ Z_i$, then we have*

$$\mathsf{GenErr}(\mathcal{F}, \{Z_i\}, \mathcal{D}) \leq 2\epsilon + \mathsf{GenErr}^{(\mathsf{Stab}_\epsilon)}(\mathcal{F}, \{Z_i\}, \mathcal{D}) = 2\epsilon + \mathsf{GenErr}(\mathcal{F}_\epsilon^0, \{Z_{i,\epsilon}^0\}, \iota_\epsilon^0(\mathcal{D})).$$

The above result says that the generalization error of our setting of interest could roughly be obtained by working with a reduced set of only the orbit representatives for points in $\mathcal{Z}$. This is in line with prior work such as [51, 20, 41]. However, note that our result is already more general and does not require that the set of transformations form a group. With this, we now need an understanding of the covering of spaces of representatives $\mathcal{Z}_\epsilon^0$ and the effect of $\mathsf{Stab}_\epsilon$ on them. The answer is evident in case $\mathsf{Stab}_\epsilon = G$, that $G$, $\mathcal{Z}^0$ are manifolds, and $\mathcal{Z} = \mathcal{Z}^0 \times G$. Since $\mathsf{ddim}$ coincides with topological dimension, and we immediately have

$$d_0 = d - \mathsf{dim}(G).$$

Intuitively, the dimensionality of $G$ can be understood as eliminating degrees of freedom from $\mathcal{Z}$, and it is this effect that improves generalization by $n^{-1/(d-\mathsf{dim}(G))} - n^{-1/d}$.

We now state a simplified form of Theorem A.3, which is sufficient for proceeding further. Our results generalize [51, Thm. 3]: we allow for non-product structure, possibly infinite sets of transformations that may not form a group, and we relax the distance deformation hypotheses for the action on $\mathcal{Z}$).

**Corollary 4.3** (of Thm. A.3)**.** *With the same notation as above, assume that for* $\mathsf{Stab}_\epsilon$ *the transformations corresponding to the action of elements of* $\mathsf{Stab}_\epsilon$ *satisfy the following for some $L > 0$, and for a choice of a set of representatives $\mathcal{Z}_\epsilon^0 \subset \mathcal{Z}$ of representatives of* $\mathsf{Stab}_\epsilon$*-orbits:*

1. *For all $z_0, z_0' \in \mathcal{Z}_\epsilon^0$ and all $g \in \mathsf{Stab}_\epsilon$ it holds that $\mathsf{d}(z_0, z_0') \leq L\,\mathsf{d}(g \cdot z_0, g \cdot z_0')$.*

2. *For all $g, g' \in \mathsf{Stab}_\epsilon$ it holds that $\mathsf{d}_G(g, g') \leq L\,\mathsf{dist}(g \cdot \mathcal{Z}_\epsilon^0, g' \cdot \mathcal{Z}_\epsilon^0)$*[6]*.*

*Then for each $\delta > 0$ there holds $\mathcal{N}(\mathcal{Z}_\epsilon^0, \delta) \leq \mathcal{N}(\mathcal{Z}, \delta/2L)/\mathcal{N}(\mathsf{Stab}_\epsilon, \delta)$.*

To be able to quantify the precise generalization benefit we need a bound on the quantity $\mathcal{N}(\mathsf{Stab}_\epsilon, \delta)$. For doing so, we assume that $\mathsf{Stab}_\epsilon$ is a finite subset of a discrete group $G$, or is a positive measure subset of a compact group $G$. As before, let $|G|$ and $\mathsf{d}_G$ denote $\sharp G$ and $\mathsf{ddim}(G)$ respectively for finite $G$. Also, denote the Hausdorff measure and dimension by $\mathsf{Vol}(G)$ and $\mathsf{dim}(G)$ for compact metric groups. Note that the minimum separation $\delta_G$ is zero for $\mathsf{dim}(G) > 0$. Then our covering bound can be expressed in the following umbrella statement:

$$\textbf{Assumption:} \quad \mathcal{N}(\mathsf{Stab}_\epsilon, \delta) \gtrsim \max\left\{1,\ \frac{|\mathsf{Stab}_\epsilon|}{(\max\{\delta, \delta_G\})^{\mathsf{d}_G}}\right\}. \tag{4.7}$$

In order to compare the above to the precise equivariance case, we later use the factor

$$\mathsf{Dens}(\epsilon) := \frac{|\mathsf{Stab}_\epsilon|}{|G|} \in (0, 1],$$

which measures the richness of $\mathsf{Stab}_\epsilon$ compared to the whole ambient group $G$, in terms of error $\epsilon$. The fact that $\mathsf{Dens}(\epsilon) > 0$ follows from our assumption that $|\mathsf{Stab}_\epsilon| > 0$. Furthermore, $\mathsf{Dens}(\epsilon)$ is always an increasing function of $\epsilon$, as follows from the definition of $\mathsf{Stab}_\epsilon$. With these considerations on coverings, we can state a general result quantifying the generalization benefit, where if we set $\epsilon = 0$, we recover the case of exact group equivariance.

**Theorem 4.4.** *Let $\epsilon > 0$ be fixed. Assume that for a given ambient group $G$ the almost stabilizers $\mathsf{Stab}_\epsilon$ satisfy assumption (4.7) and that the assumptions of Corollary 4.3 hold for some finite $L = L_\epsilon > 0$. Then with the notations of Proposition 4.1 we have with probability $\geq 1 - \delta$*

$$\mathsf{GenErr}(\mathcal{F}, \{Z_i\}, \mathcal{D}) \lesssim n^{-1/2}\sqrt{\|\mathcal{F}\|_\infty \log(2/\delta)} + 2\epsilon + (E_\epsilon),$$

*where*

$$(E_\epsilon) := \begin{cases} \frac{4^{d_0} d_0}{d_0 - 2} \delta_G^{-d_0/2+1} \left(\frac{(2L_\epsilon)^d D^d}{|\mathsf{Stab}_\epsilon|n}\right)^{1/2} & \text{if } G \text{ is finite and } (2L_\epsilon)^d D^d < |\mathsf{Stab}_\epsilon|n\,\delta_G^{d_0}, \\[2ex] \frac{4^{d_0} d_0}{d_0 - 2}\left(\frac{(2L_\epsilon)^d D^d}{|\mathsf{Stab}_\epsilon|n}\right)^{1/d_0} & \text{else.} \end{cases}$$

The interpretation of the above terms is direct: the term $2\epsilon$ is the effect of approximate equivariance, and the term $(E_\epsilon)$ includes the dependence on $|\mathsf{Stab}_\epsilon|$ and thus is relevant to the study of partial equivariance. In general the Lipschitz bound $L_\epsilon$ is increasing in $\epsilon$ as well, since $\mathsf{Stab}_\epsilon$ includes more elements of $G$ as $\epsilon$ increases, and we can expect that actions of elements generating higher equivariance error in general have higher oscillations.

## 5  Approximation error bounds under approximate data equivariance

In the introduction, we stated that we aim to validate the intuition that that model is best whose symmetry aligns with that of the data. So far we have only given bounds on the generalization error. Note that our bounds hold for any data distribution: in particular, the bounds are not affected by whether the data distribution is $G$-equivariant or not. However,

---

[6]Here $\mathsf{dist}(A, B) := \min\{\mathsf{d}(a, b) :\ a \in A, b \in B\}$ denotes the distance between sets, induced by $\mathsf{d}$.

the fact that data may have fewer symmetries than the ones introduced in the model will have a controlled effect on worsening the approximation error, as described in this section.

However, controlling the approximation error has an additional actor in the fray to complicate matters: the distinguishing power of the loss function. Indeed, having a degenerate loss function with a large minimum set will not distinguish whether the model is fit to the data distribution or not. Thus, lowering the discrimination power of the loss has the same effect as increasing the function space for which we are testing approximation error.

Assuming at first that $\widetilde{\mathcal{F}}$ is composed of $G$-equivariant functions, we compare its approximation error to the one of the set $\mathcal{M}$ of all measurable functions $m : \mathcal{X} \to \mathcal{Y}$. Recall that $f(Z) = \ell(\widetilde{f}(X), Y)$ is a positive function for all $\widetilde{f} \in \widetilde{\mathcal{F}}$, thus, denoting by $\mathcal{M}' := \{F(x,y) = \ell(m(x), y), m \in \mathcal{M}\}$, the approximation error gap can be defined and bounded as follows (where $F^* \in \mathcal{M}'$ is a function realizing the first minimum in the second line below):

$$
\begin{aligned}
\mathsf{AppGap}(\mathcal{F}, \mathcal{D}, ) \quad := \quad & \mathsf{AppErr}(\mathcal{F}, \mathcal{D}) - \mathsf{AppErr}(\mathcal{M}', \mathcal{D}) := \min_{f \in \mathcal{F}} \mathbb{E}[f(Z)] - \min_{F \in \mathcal{M}'} \mathbb{E}[F(Z)] \\
\geq \quad & \min_{F \in \mathcal{M}'} \mathbb{E}[\mathbb{E}_g[F(g \cdot Z)]] - \min_{F \in \mathcal{M}'} \mathbb{E}[F(Z)] \geq \mathbb{E}[\mathbb{E}_g[F^*(g \cdot Z)]] - \mathbb{E}[F^*(Z)] \\
\geq \quad & \min_{F \in \mathcal{M}'} \mathbb{E}[\mathbb{E}_g[F(g \cdot Z)] - F(Z)].
\end{aligned}
$$

With applications to neural networks in mind, we will work with the following simplifying assumptions:

1. The loss function quantitatively detects whether the label is correct, i.e. we have $\ell(y, y') = 0$ if $y = y'$, and $\ell(y, y') \geq \varphi(\mathsf{d}_{\mathcal{Y}}(y, y'))$ for a strictly convex $\varphi$ with $\varphi(0) = 0$, where $\mathsf{d}_{\mathcal{Y}}$ is the distance on $\mathcal{Y}$. This assumption seems reasonable for use in applications, where objective functions with strict convexity properties are commonly used, as they help for the convergence of optimization algorithms.
2. The minimization across all measurable functions $\widetilde{\mathcal{F}} = \mathcal{M}$ produces a zero-risk solution, i.e. $\min_{F \in \mathcal{M}'} \mathbb{E}[F(Z)] = 0$ is achieved by a measurable function $y^* : \mathcal{X} \to \mathcal{Y}$. By assumption 1, this implies that $\mathcal{D}$ is concentrated on the graph of $y^*$. This assumption corresponds to saying that the learning task is in principle deterministically solvable. Under this assumption, the task becomes to approximate to high accuracy the (unknown) solution $y^*$.

With the above hypotheses we have

$$
\mathsf{AppGap}(\mathcal{F}, \mathcal{D}) = \mathsf{AppErr}(\mathcal{F}, \mathcal{D}) \geq \min_{F \in \mathcal{M}'} \mathbb{E}_Z[\mathbb{E}_g[F(g \cdot Z)]].
$$

As a slight simplification of Assumption 1 for ease of presentation, we will simply take $\mathcal{Y} = \mathbb{R}^d$ and $\ell(y, y') = \mathsf{d}_{\mathcal{Y}}(y, y')^2$ below. Note that Assumption 1 directly reduces to this case, as a strictly convex $\varphi(t)$ with $\varphi(0) = 0$ as in Assumption 1 is automatically bounded below by a multiple of $t^2$.

Now, with the aim to capture the mixed effects of approximate and partial equivariance, we introduce the following set of model classes. For $\epsilon \geq 0, \lambda \in (0, 1]$:

$$
\mathcal{C}_{\epsilon, \lambda} := \left\{ \mathcal{F} \subset \mathcal{M}' : \ \mathsf{Dens}(\epsilon) = \frac{|\mathsf{Stab}_\epsilon(\mathcal{F})|}{|G|} \geq \lambda \right\}.
$$

In order to establish some intuition, note that $\mathsf{Stab}_\epsilon(\mathcal{F})$ is increasing in $\epsilon$ and as a consequence $\mathcal{C}_{\lambda, \epsilon}$ is also increasing in $\epsilon$. Directly from the definition one finds that $\mathsf{Stab}_0(\mathcal{F})$ is necessarily a subgroup of $G$, and thus we allow $\epsilon > 0$, which allows more general sets of symmetries than just subgroups of $G$. The most interesting parameter in the above definition of $\mathcal{C}_{\epsilon, \lambda}$ is $\lambda$, which actually bounds from below the "amount of prescribed symmetries" for our models. In the fully equivariant case $\epsilon = 0, \lambda = 1$, we have the simple description $\mathcal{C}_{0,1} = \{\mathcal{F} : \mathcal{F} \subseteq (\mathcal{M}^{(G)})'\}$, whose maximal element is $(\mathcal{M}^{(G)})'$. However in general $\mathcal{C}_{\epsilon, \lambda}$ will not have a unique maximal element for $\lambda < 1$: this is due to the fact that two different elements $\mathcal{F}_1 \neq \mathcal{F}_2 \in \mathcal{C}_{\epsilon, \lambda}$ may have incomparable stabilizers, so that $|\mathsf{Stab}_\epsilon(\mathcal{F}_1) \setminus \mathsf{Stab}_\epsilon(\mathcal{F}_2)| > 0$ and $|\mathsf{Stab}_\epsilon(\mathcal{F}_2) \setminus \mathsf{Stab}_\epsilon(\mathcal{F}_1)| > 0$, and thus $\mathcal{F}_1 \cup \mathcal{F}_2 \notin \mathcal{C}_{\epsilon, \lambda}$. Our main result towards our approximation error bound is the following.

**Proposition 5.1.** *Assume that $(\mathcal{Y}, \mathsf{d}_{\mathcal{Y}})$ is a compact metric space and consider $\ell(y, y') = \mathsf{d}_{\mathcal{Y}}(y, y')^2$. Fix $\epsilon \geq 0$. Then for each $\lambda \in (0, 1]$ there exists an element $\mathcal{F} \in \mathcal{C}_{\epsilon, \lambda}$ and a measurable selection $\iota_{\epsilon}^0 : \mathcal{X} \to \mathcal{X}_{\epsilon}^0$ of representatives of orbits of $\mathcal{X}$ under the action of $\mathsf{Stab}_{\epsilon}(\mathcal{F})$, such that if $X_{\epsilon}^0, g$ are random variables obtained from $X \sim \mathcal{D}$ by imposing $X_{\epsilon}^0 := \iota_{\epsilon}^0 \circ X$ and $X = g \cdot X_{\epsilon}^0$, then we have*

$$\mathsf{AppErr}(\mathcal{F}, \mathcal{D}) \leq \mathbb{E}_{X_{\epsilon}^0} \min_y \mathbb{E}_{g | X_{\epsilon}^0} \left[ \left( \max \left\{ \mathsf{d}_{\mathcal{Y}} \left( g \cdot y, \ y^*(g \cdot X_{\epsilon}^0) \right) - \epsilon, \ 0 \right\} \right)^2 \right].$$

In the above, as $\epsilon$ increases, we see a decrease in the approximation gap, since in this case model classes $\mathcal{F}$ allow more freedom for approximating $y^*$ more accurately. On the other hand, the effect of parameter $\lambda$ is somewhat more subtle, and it has to do with the fact that as $\lambda$ decreases, $\mathsf{Stab}_{\epsilon}(\mathcal{F})$ for $\mathcal{F} \in \mathcal{C}_{\epsilon, \lambda}$ can become smaller and the allowed oscillations of $y^*(g \cdot X_{\epsilon}^0)$ are more and more controlled.

Now to bound the approximation error above using the quantities that we have been working with, we will use the following version of an **isodiametric inequality** on metric groups $G$. As before we assume that $G$ has Hausdorff dimension $\mathsf{dim}(G) > 0$ which coincides with the doubling dimension, or is discrete. By $|X|$ we denote the Hausdorff measure of $X \subseteq G$ if $\mathsf{dim}(G) > 0$ or the cardinality of $X$ if $G$ is finite. Then there exists an isodiametric constant $C_G$ depending only on $G$ such that for all $X \subseteq G$ it holds that

$$\frac{|X|}{|G|} \geq \lambda \quad \Rightarrow \quad \mathsf{diam}(X) \geq C_G \lambda^{1/\mathsf{ddim}(G)}. \tag{5.1}$$

The above bound can be obtained directly from the doubling property of $G$, applied to a cover of $X$ via a single ball of radius $\mathsf{diam}(X)$, which can be refined to obtain better and better approximations of $|X|$. We can now control the bound from Proposition 5.1 as follows:

**Theorem 5.2.** *Assume that $(\mathcal{Y}, \mathsf{d}_{\mathcal{Y}})$ is a compact metric space and $\ell(y, y') = \mathsf{d}_{\mathcal{Y}}(y, y')^2$. Let $G$ be a compact Lie group or a finite group, with distance $\mathsf{d}_G$. Let $\mathsf{ddim}(G)$ be the doubling dimension of $G$, and assume that for all $g \in G$ such that the action of $g$ over $\mathcal{X}$ is defined, it holds that $\mathsf{d}(g \cdot x, g' \cdot x) \leq L' \mathsf{d}_G(g, g')$. Then, there exists a constant $C_G$ depending only on $G$ such that for all $\lambda \in (0, 1]$ and $\epsilon > 0$, there exists an element $\mathcal{F} \in \mathcal{C}_{\epsilon, \lambda}$ where for any data distribution $\mathcal{D}$, $y^*$ can be chosen Lipschitz with constant $\mathsf{Lip}(y^*)$. We have*

$$\mathsf{AppErr}(\mathcal{F}, \mathcal{D}) \leq \left( \max \left\{ C_G \ L' \ (1 + \mathsf{Lip}(y^*)) \lambda^{1/\mathsf{ddim}(G)} - \epsilon, \ 0 \right\} \right)^2.$$

We note that, given our proof strategy of Proposition 5.1, we conjecture that actually the above bound is sharp up to a constant factor: $\min_{\mathcal{F} \in \mathcal{C}_{\epsilon, \lambda}} \mathsf{AppErr}(\mathcal{F}, \mathcal{D})$ may have a lower bound comparable to the above upper bound. The goal of the above result is to formalize the property that the approximation error guarantees of $\mathcal{F}$ will tend to grow as the amount of imposed symmetries grow, as measured by $\lambda$.

## 6 Discussion: Imposing optimal equivariance

We have so far studied bounds on the generalization and the approximation errors, each of which take a very natural form in terms of properties of the space of orbit representatives (and thus the underlying $G$). One of them depends on the data distribution, while the other doesn't. Using them we can thus state a result that quantifies the performance error, or gives the generalization-approximation trade-off, of a model based on the data distribution (and their respective symmetries). Define the performance error of a model class $\mathcal{F} \in \mathcal{C}_{\epsilon, \lambda}$ over an i.i.d. sample $\{Z_i\}_{i=1}^n$, $Z_i \sim \mathcal{D}$ as

$$\mathsf{PerfErr}(\mathcal{F}, \{Z_i\}, \mathcal{D}) := \mathsf{GenErr}(\mathcal{F}, \{Z_i\}, \mathcal{D}) + \mathsf{AppErr}(\mathcal{F}, \mathcal{D}).$$

Combining Theorems 4.4 and 5.2, we get the following:

**Theorem 6.1.** *Under the assumptions of Theorems 4.4 and 5.2, we have that, assuming all the function classes are restricted to functions with values in $[-M, M]$, then there exist*

*values $C_1, C_2, C_3 >$ depending only on $d_0, (2LD)^d/|G|, \delta_G$ and $C = C_G L'(1 + \mathsf{Lip}(y^*))$ such that with probability at least $1 - \delta$,*

$$\mathsf{PerfErr}(\mathcal{F}, \{Z_i\}, \mathcal{D}) \lesssim n^{-1/2}\sqrt{M\log(2/\delta)} + 2\epsilon + (C\lambda^{1/\mathsf{d}_G} - \epsilon)_+^2 + \begin{cases} C_1 \dfrac{1}{(n\lambda)^{1/2}} & \text{if } n\lambda \geq C_3, \\[2ex] C_2 \dfrac{1}{(n\ \lambda)^{1/d_0}} & \text{if } n\ \lambda < C_3. \end{cases}$$

Note that the above bounds are not informative as $\lambda \to 0$ for the continuous group case, which corresponds to the second option in Theorem 6.1. This can be sourced back to the form of Assumption (4.7), in which the covering number on the left is always $\geq 1$, whereas in the continuous group case, the right-hand side may be arbitrarily small.

Thus the result is only relevant for $\lambda$ away from 0. If we fix $\epsilon = 0$ and we optimize over $\lambda \in (0,1]$ in the above bound, we find a unique minimum $\lambda^*$ for the error bound, specifying the optimum size of $\mathsf{Stab}_\epsilon$ for the corresponding case (see the Appendix for more details). This validates the intuition stated at the onset that for the best model its symmetries must align with that of the data. As an illustration, for $C, C_1, C_2 = 0.04$ and $C_3 = 0.01$ with $n = 1000000$, the error trade-off is shown on the figure on the right.

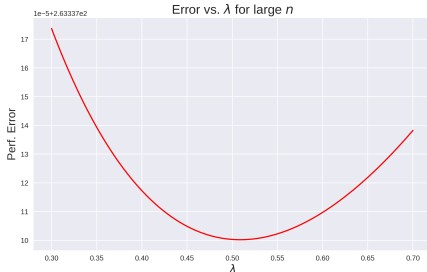

Error versus lambda for large $n$ for fixed values of $C, C_1, C_2, C_3$.

# 7 Concluding Remarks

In this paper, we have provided general quantitative bounds showing that machine learning models with task-pertinent symmetries improve generalization. Our results do not require the symmetries to be a group and can work with partial/approximate equivariance. We also presented a general result which confirms the prevalent intuition that if the symmetry of a model is mis-specified w.r.t to the data symmetry, its performance will suffer. We now indicate some important future research directions, that correspond to limitations of our work:

- (*Model-specific bounds*) While in Theorem 6.1 we obtain the existence of an optimal amout $\lambda^*$ without hopes to be sharp in such generality, we may expect that if we restrict to specific tasks, the bounds can become much tighter, and if so, the value $\lambda^*$ of optimal amount of symmetries can become interpretable and give insights into the structure of the learning tasks.

- (*Tighter bounds beyond classical PAC theory*) For the sake of making the basic principles at work more transparent, we based our main result on long-known classical results. However, tighter data-dependent or probabilistic bounds would be very useful for getting a sharper value for the optimal value of symmetries $\lambda^*$ via a stronger version of Theorem 6.1.

- (*Beyond controlled doubling dimension*) Our focus here is on groups $G$ of controlled doubling dimension. This includes compact and nilpotent Lie groups, and discrete groups of polynomial growth, but the doubling dimension is not controlled for notable important cases, such as for the permutation group $S_n$ (cf. section 3 of [17]) or for the group $(\mathbb{Z}/2\mathbb{Z})^n$. In order to cover these cases, it will be interesting to build analogues of our main result based on other group complexity bounds, beyond the doubling dimension case.

## Acknowledgments and Disclosure of Funding

The authors thank Aryeh Kontorovich for readily answering questions (and supplying references) about Kolmogorov-Tikhomirov, and Bryn Elesedy for graciously answering queries and providing a copy of his dissertation before it appeared online. MP thanks the Centro Nacional de Inteligencia Artificial in Chile for support, as well as Fondecyt Regular grant number 1210462 entitled "Rigidity, stability and uniformity for large point configurations".

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

# A   Proofs

## A.1   Proofs of results from Section 4

### A.1.1   Proof of Proposition 4.1

*Proof.* Reformulating (4.1) we find that with probability at least $1 - \delta$ it holds that:

$$\sup_{f \in \mathcal{F}} |(P - P_n)f| \leq \mathcal{R}(\mathcal{F}_Z) + \frac{\sqrt{2\|\mathcal{F}\|_\infty \log(2/\delta)}}{\sqrt{n}}. \tag{A.1}$$

Next, we compound the bounds for $\mathcal{R}(\mathcal{F}_Z)$. The optimum $\alpha > 0$ in the first line of (4.2) is the one for which $\frac{n}{9} = \ln \mathcal{N}(\mathcal{F}, \alpha, \|\cdot\|_\infty)$. Keeping in mind (4.6), we choose $\alpha = \mathsf{diam}(\mathcal{Z})n^{-\frac{1}{\mathsf{ddim}\mathcal{Z}}}$ in (4.2), which gives (abbreviating $d = \mathsf{ddim}\mathcal{Z}, D = \mathsf{diam}\mathcal{Z}$)

$$\begin{aligned}
\mathcal{R}(\mathcal{F}_Z) &\lesssim D/n^{1/d} + n^{-1/2} \int_{Dn^{-1/d}}^\infty (D/t)^{d/2}dt = D/n^{1/d} + (D/n)^{1/2} \int_{n^{-1/d}}^\infty \tau^{-d/2}d\tau \\
&= \frac{D}{n^{1/d}}\left(1 + \frac{2}{d-2}\right).
\end{aligned}$$

Now the last equation and equation (A.1) yield the claim. $\qquad\square$

### A.1.2   Proof of Proposition 4.2

We start with a preliminary result under hypotheses of strict equivariance. In this case, we are able to use a change of variables to reduce the generalization error formula to an equivalent one depending only on a measurable choice of $G$-orbit representatives of elements from $\mathcal{Z}$:

**Proposition A.1.** *Let $\mathcal{F}$ be a set of $G$-invariant functions, and let $\mathcal{Z}^0 \subset \mathcal{Z}$ be a choice of $G$-orbit representatives for points in $\mathcal{Z}$, such that $\iota^0 : \mathcal{Z} \to \mathcal{Z}^0$ associating to each $z \in \mathcal{Z}$ its orbit representative $z^0$, is Borel measurable. Let $\mathcal{F}^0 := \{f|_{\mathcal{Z}^0} : f \in \mathcal{F}\}$ and denote by $\iota^0(\mathcal{D})$ the image measure of $\mathcal{D}$. Then for each $n \in \mathbb{N}$ if $\{Z_i\}_{i=1}^n$ are i.i.d. samples with $Z_i \sim \mathcal{D}$ and $Z_i^0 := \iota^0 \circ Z_i$, we have*

$$\mathsf{GenErr}(\mathcal{F}, \{Z_i\}, \mathcal{D}) = \mathsf{GenErr}^{(G)}(\mathcal{F}, \{Z_i\}, \mathcal{D}) = \mathsf{GenErr}(\mathcal{F}^0, \{Z_i^0\}, \iota^0(\mathcal{D})).$$

*Proof.* For the first equality, we use the definition of $\mathsf{GenErr}$ and the change of variable formula (3.1) and the fact that $G$-invariant functions $f$ satisfy $f(Z) = \mathbb{E}_g[f(g \cdot Z)]$. For the second equality, note that by hypothesis, for each $f \in \mathcal{F}$ we have $f(z) = f(g \cdot z)$ for all $g \in G, z \in \mathcal{Z}$, in particular $f(z) = f(\iota^0(z))$ and we conclude by a change of variable by the map $\iota^0$ in the expectations from the definition of $\mathsf{GenErr}^{(G)}$. $\qquad\square$

Now the proof Proposition 4.2 combines the above idea with a simple extra step:

*Proof of Proposition 4.2:* The proof uses the triangular inequality. For $f \in \mathcal{F}$ and $g \in \mathsf{Stab}_\epsilon$, we have:

$$|Pf - P_nf| = \left|\mathbb{E}[f(Z)] - \frac{1}{n}\sum_{i=1}^n f(Z_i)\right| \leq \left|\mathbb{E}[g \cdot f(Z)] - \frac{1}{n}\sum_{i=1}^n g \cdot f(Z_i)\right| + 2\|g \cdot f - f\|_\infty.$$

By averaging over $g \in \mathsf{Stab}_\epsilon$, we obtain the inequality in the statement of the proposition. The equality follows by a change of variable via map $\iota_\epsilon^0$, exactly as in the strategy of Proposition A.1. $\qquad\square$

### A.1.3   Proof of Corollary 4.3 and of the more general result of Theorem A.3

In order to make the treatment better digestible, we first consider the intuitively simpler case of strict equivariance, and then describe how to extend it to the more general case of approximate and partial equivariance. In this case, if we restrict our equivariant functions to only the space of orbit representatives $\mathcal{Z}^0$, the dimension counts from classical generalization bounds of Proposition 4.1 improve as follows:

**Corollary A.2.** *Assume that $\mathcal{F}$ is composed of $G$-invariant functions and that $d_0 := \mathsf{ddim}(\mathcal{Z}^0) > 2$. Also, denote $D_0 := \mathsf{diam}(\mathcal{Z}^0)$. With the same notation as in Proposition A.1 and with the hypotheses of Proposition 4.1, for any probability distribution $\mathcal{D}$ over $\mathcal{Z}$, the following holds with probability at least $1 - \delta$:*

$$\mathsf{GenErr}(\mathcal{F}, \{Z_i\}, \mathcal{D}) \lesssim \frac{d_0}{d_0 - 2} \left( \frac{D_0^{d_0}}{n} \right)^{1/d_0} + n^{-1/2} \sqrt{\|\mathcal{F}\|_\infty \log(2/\delta)}.$$

*Proof.* Due to Proposition A.1, we only need to bound $\mathsf{GenErr}(\mathcal{F}^0, \{Z_i^0\}, \iota^0(\mathcal{D}))$. Thus it suffices to apply Proposition 4.1 for the above function. We note that $\|\mathcal{F}^0\|_\infty \leq \|\mathcal{F}\|_\infty$ to conclude. $\square$

The drawback of the above Corollary, is that it leaves open the question of *how to actually bound* the diameter and dimension of $\mathcal{Z}^0$, on which we do not have direct control. The next steps we take consist precisely in translating the information from properties of $G$ to relevant properties of $\mathcal{Z}^0$.

A first, simpler, approach could be the following. Under the reasonable assumption that $\mathcal{Z}, \mathcal{Z}^0$ have diameter greater than 1, the leading term on the left in Corollary A.2 is $n^{-1/d_0}$. Thus the optimal choices of $\mathcal{Z}^0$ are those which minimize the doubling dimension $d_0 = \mathsf{ddim}(\mathcal{Z}^0)$ amongst sets of representatives of $G$-orbits. This is a weak regularity assumption, implying that we want $\mathcal{Z}^0$ to not oscillate wildly. The effect of $G$ on coverings is evident in case $G$, $\mathcal{Z}^0$ are manifolds, and $\mathcal{Z} = \mathcal{Z}^0 \times G$ (see (A.3) for the strictly equivariant case, and the more general (A.5) for the general case). Since $\mathsf{ddim}$ coincides with topological dimension, we immediately have

$$d_0 = d - \mathsf{dim}(G).$$

Intuitively, the dimensionality of $G$ can be understood as eliminating degrees of freedom from $\mathcal{Z}$, and it is this effect that improves generalization by $n^{-1/(d - \mathsf{dim}(G))} - n^{-1/d}$.

In order to include more general situations, we now describe a second, more in-depth approach. We take a step back and rather than addressing direct diameter and dimension bounds for $\mathcal{Z}^0$, we go "back to the source" of Proposition 4.1. We update the bounds on covering numbers of $\mathcal{Z}^0$, directly in terms of the $G$-action and of $\mathcal{Z}$. The ensuing framework is robust enough to later include, after a few adjustments, also the cases of partial and approximate equivariance. Here is our fundamental bound, which generalizes and extends [51, Thm.3].

**Theorem A.3.** *Assume that $\mathcal{Z}$ is a metric space with distance $d$ and $S \subset G$ is a subset of a metric group $G$ consisting of transformations $g : \mathcal{Z} \to \mathcal{Z}$ (with action denoted $g \cdot z := g(z)$) for which there exists a choice of $S$-orbit representatives $\mathcal{Z}_0 \subset \mathcal{Z}$ and a distance function $\mathsf{d}$ on $S$ satisfying the following for $L, L' \in (0, +\infty]$ (with the conventions $1/+\infty := 0$ and $\mathcal{N}(X, 0) := +\infty, \mathcal{N}(X, +\infty) := 0$):*

1. *For all $z_0, z_0' \in \mathcal{Z}_0$ and all $g \in S$ it holds that $\frac{1}{L} \mathsf{d}(z_0, z_0') \leq \mathsf{d}(g \cdot z_0, g \cdot z_0') \leq L' \mathsf{d}(z_0, z_0')$.*

2. *For all $g, g' \in S$ and $z_0, z_0' \in \mathcal{Z}_0$ it holds that $\frac{1}{L} \mathsf{d}_G(g, g') \leq \mathsf{d}(g \cdot z_0, g' \cdot z_0') \leq L' \mathsf{d}_G(g, g')$.*

*Then for each $\delta' > 0$ the following holds*

$$\mathcal{N}(\mathcal{Z}_0, 2\delta' L) \mathcal{N}(S, 2\delta' L) \leq \mathcal{N}(\mathcal{Z}, \delta') \leq \mathcal{N}(\mathcal{Z}_0, \delta'/2L') \mathcal{N}(S, \delta'/2L'). \tag{A.2}$$

Before the proof, we observe how Corollary 4.3 can be recovered using the choice $L > 0, L' = +\infty$ in Theorem A.3:

*Proof of Corollary 4.3:* We apply Theorem A.3 to $S = \mathsf{Stab}_\epsilon$ and $\mathcal{Z}_0 = \mathcal{Z}_\epsilon^0$ as in the corollary statement. Then we take $\delta = 2L\delta'$ in the conclusion (A.2), and consider only the lower bound inequality, which directly gives the claim of the corollary. $\square$

*Proof of Theorem A.3:* In this proof, we will denote a minimal $\alpha$-ball cover of a metric space $X$ by $X_\alpha$.

Note that we are not assuming $G$ to be a group, but due to lower bound in property 1. it follows that $z_0 \mapsto g \cdot z_0$ is injective (for $z_0 \neq z_0'$ we have $\mathsf{d}(z_0, z_0') > 0$ and thus $g \cdot z_0 \neq g \cdot z_0'$), and when below we write "$g^{-1}$" this has to be interpreted as the inverse of the $g$-action, restricted to its image.

Further, note that the case when one of $L, L'$ is $+\infty$, corresponds to removing the part of the assumptions (and of the conclusions) involving that value, thus we only consider the case of finite $L'$ and finite $L$.

On fixing an arbitrary point $z \in \mathcal{Z}$, we can write $z = g \cdot z_0$ for a suitable $g \in G, z_0 \in \mathcal{Z}^0$. Let $\eta := \delta'/2L'$. For fixed covers $\mathcal{Z}_\eta^0, G_\eta$, there exists a point $z_0' \in \mathcal{Z}_\eta^0$ with $\mathsf{d}(z_0', z_0) < \eta$ and $g' \in G_\eta$ with $\mathsf{d}_G(g', g) < \eta$. Thus by property 1. we have $\mathsf{d}(g \cdot z_0', z) < L'\eta = \epsilon/2$ and by property 2. we have $\mathsf{d}(g' \cdot z_0', g \cdot z_0') < L'\eta = \delta'/2$. By the triangle inequality, $\mathsf{d}(g' \cdot z_0', z) < \delta'$ and thus $G_\eta \cdot \mathcal{Z}_\eta^0$ is an $\delta'$-cover of $\mathcal{Z}$. Thus we have

$$\mathcal{N}(\mathcal{Z}, \delta') \leq \#G_{\delta'/2L'} \, \#\mathcal{Z}_{\delta'/2L'}^0,$$

optimizing over the cardinalities on the right hand side yields the second inequality in (A.2).

Now consider an $\delta'$-cover $\mathcal{Z}_{\delta'}$ of $\mathcal{Z}$, and for $\eta = 2\delta L$ consider an $\eta$-cover $G_\eta$ of $G$. We find that for each $z \in \mathcal{Z}_{\delta'}$, there exists at most one $g \in G_\eta$ such that $\mathsf{dist}(z, g \cdot \mathcal{Z}_0) < \eta/2L = \delta'$. Notice that if this were false, we could use the triangle inequality and contradict property 2. in the statement. For each $g \in G_\eta$ denote $Z_g$ the set of such points $z \in \mathcal{Z}_{\delta'}$ such that there exists $x \in g \cdot \mathcal{Z}^0$, and assign exactly one such $x = x(z)$ to each $z$, forming a set $X_g$ of all such $x(z)$. Any other point $x' \in g \cdot \mathcal{Z}^0$ such that $\mathsf{d}(x', z) < \delta'$ then satisfies $\mathsf{d}(x', x) < 2\delta'$ by triangle inequality, and thus $X_g$ is a $2\delta'$-cover of $g \cdot \mathcal{Z}_0$. If for $g \cdot z_0 \in g \cdot \mathcal{X}_0$ the point $x \in g \cdot \mathcal{X}_0$ satisfies $\mathsf{d}(g \cdot z_0, x) < 2\delta'$, then by property 1. in the statement we have $\mathsf{d}(z_0, g^{-1} \cdot x) < 2\delta'L$, and thus $g^{-1} \cdot X_g$ is a $2\delta'L$-cover of $\mathcal{Z}_0$, having the same cardinality as $Z_g$. We then compute as follows, proving the first inequality in (A.2):

$$\mathcal{N}(\mathcal{Z}, \delta') \geq \sum_{g \in G_\eta} \#Z_g = \sum_{g \in G_\eta} \#(g^{-1} \cdot X_g) \geq \mathcal{N}(G, 2\delta'L)\mathcal{N}(\mathcal{Z}_0, 2\delta'L).$$

$\square$

### A.1.4   Proof of Theorem 4.4

As before, we focus again first on the exact equivariance case, where Theorem A.4 is the direct analogue to (or special case of) Theorem 4.4.

Under the hypotheses of Theorem A.3 on the $G$-action, we directly obtain the following, for the strictly equivariant case:

$$\mathcal{N}(\mathcal{Z}_0, \delta) \leq \frac{\mathcal{N}(\mathcal{Z}, \delta/2L)}{\mathcal{N}(G, \delta)}. \tag{A.3}$$

We next impose that for $\delta \lesssim \mathsf{diam}(G)$ the group $G$ satisfies the natural "volume growth" assumption, where for compact groups $\mathsf{Vol}(G)$ is its $\mathsf{dim}(G)$-dimensional Hausdorff measure and $\mathsf{dim}(G)$ is the usual Hausdorff dimension, and for finite groups we use minimum separation notation $\delta_G > 0$ as defined in (4.4):

$$\textbf{Assumption:} \quad \mathcal{N}(G, \delta) \gtrsim \begin{cases} \#G/(\max\{\delta, \delta_G\})^{\mathsf{ddim}(G)}, & \text{if } G \text{ finite,} \\ \mathsf{Vol}(G)/\delta^{\mathsf{dim}G}, & \text{if } \mathsf{dim}G > 0. \end{cases} \tag{A.4}$$

Similarly to Proposition 4.1, we then get the following, in which the leading term in the bound has exponent figuring $d_0 = \mathsf{ddim}(\mathcal{Z}) - \mathsf{dim}(G)$. Recall that $\mathsf{dim}(G) = 0$ for finite groups, thus the distinction can be made directly in terms of the dimension of $G$.

**Theorem A.4.** *Let $\delta > 0$ be fixed. Assume that for a given ambient group $G$ the group $G$ satisfies assumption (A.4) and that its action satisfies the the assumptions 1. and 2. of Theorem A.3 for some finite $L > 0$ and $L' = +\infty$. We denote $d_G = \mathsf{ddim}(G)$ if $G$ is a discrete group and $d_G = \mathsf{dim}(G)$ if $G$ is compact and non-discrete, and $d = \mathsf{ddim}(\mathcal{Z})$. Furthher assume $d_0 := d - d_G > 2$. Furthermore, set $|G| := \mathsf{Vol}(G)$ if $\mathsf{dim}G > 0$ and $|G| := \#G$ for finite $G$. Then with the notations of Proposition 4.1 we have with probability $\geq 1 - \delta$*

$$\mathsf{GenErr}(\mathcal{F}, \{Z_i\}, \mathcal{D}) \lesssim n^{-1/2}\sqrt{\|\mathcal{F}\|_\infty \log(2/\delta)} + (E),$$

*where*

$$(E) := \begin{cases} \frac{d_0}{d_0-2} \delta_G^{-d_0/2+1} \left( \frac{(2L)^d D^d}{|G|n} \right)^{1/2} & \text{if } G \text{ is finite and } (2L)^d D^d < |G|n\ \delta_G^{d_0}, \\[2ex] \frac{d_0}{d_0-2} \left( \frac{(2L)^d D^d}{|G|n} \right)^{1/d_0} & \text{otherwise.} \end{cases}$$

*Proof.* We follow the same computation as Proposition 4.1, but use Proposition A.1 in order to reduce to restrictions of functions to $\mathcal{Z}^0$. In this case, using (A.3) and assumption (A.4), and with notation as in our statement, we will have:

$$\mathcal{N}(\mathcal{Z}_0, t) \lesssim \frac{(2L)^d D^d}{|G|} \max\{\delta_G, t\}^{-d_0},$$

where we have $\delta_G = 0$ for $\mathsf{dim}\, G > 0$. We set $C := \frac{(2L)^d D^d}{|G|}$ for simplicity of notation. In case $C\delta_G^{d_0} < 1$ (which includes the case $\mathsf{dim}\, G > 0$), we take $\alpha = (C/n)^{1/d_0}$ in the Dudley integral (4.2) and find

$$\mathcal{R}(\mathcal{F}_{Z^0}) \lesssim \alpha + n^{-1/2} \int_\alpha^\infty \sqrt{Ct^{-d_0}} dt,$$

from which the proof goes exactly as in Proposition 4.1, with $C$ replacing $D^d$, and we get the second option for the value of $(E)$ as given in our statement. In case $C\delta_G^{d_0} < 1$ instead we take $\alpha = 0$ and our above bound for $\mathcal{N}(\mathcal{Z}_0, t)$ plugged into (4.2) (recalling the notation for $C$):

$$\mathcal{R}(\mathcal{F}_{Z^0}) \lesssim \int_0^\infty \sqrt{C \max\{\delta_G, t\}^{-d_0}} dt = \delta_G^{-d_0/2+1} \sqrt{C/n} + \sqrt{C/n} \int_{\delta_G}^\infty t^{-d_0/2} dt,$$

from which the second case of the value of $(E)$ follows by direct computation. $\qquad\square$

Now the proof of Theorem 4.4 proceeds in exactly the same manner as the above. Below we explain the required adaptations:

*Proof of Theorem 4.4:* The following updates are the principal adaptations required for the above proof of Theorem A.4:

- The role of $G$ should be replaced by $\mathsf{Stab}_\epsilon$, except for the fact that parameters $\delta_G, d_G$ remain unchanged (i.e. we use their values corresponding to "ambient" group $G$ rather than those for $\mathsf{Stab}_\epsilon$).

- The $G$-orbit representative set $\mathcal{Z}^0$ then should be replaced by representatives $\mathcal{Z}_\epsilon^0$ for orbits of $\mathsf{Stab}_\epsilon$.

With these changes, assumption (A.4) implies its more general version, assumption (4.7). Indeed, $|G|$ equals $\#G$ for finite $G$ and $\mathsf{Vol}(G)$ for compact $G$, and $\delta_G > 0$ only in the first case. Furthermore, we have $\delta_{\mathsf{Stab}_\epsilon} \geq \delta_G$ as a direct consequence of $\mathsf{Stab}_\epsilon \subseteq G$.

We observe that Corollary 4.3 (which also is obtained from Theorem A.3 with the above two main substitutions) directly gives the version of Theorem (A.3) required to get the correct replacement of (A.3) under our initially declared two substitutions. We get:

$$\mathcal{N}(\mathcal{Z}_\epsilon^0, \delta) \leq \frac{\mathcal{N}(\mathcal{Z}, \delta/2L)}{\mathcal{N}(\mathsf{Stab}_\epsilon, \delta)}. \tag{A.5}$$

With the above changes, the proof follows by exactly the same steps as in the above proof of Theorem A.4. $\qquad\square$

*Remark* A.5. Note that, as might be evident from the last proof, we could have introduced new more precise parameters to keep track of dimensionality and minimum separation for $\mathsf{Stab}_\epsilon$ rather than formulating assumption (4.7) in terms of $d_G, \delta_G$. This is justified for the aims of this work. Indeed, all the main situations of interest to us are those in which $\mathsf{Stab}_\epsilon$ is a "large" subset of $G$, i.e. it has dimension $d_G$, and in all our examples for finite groups $\delta_G > 0$, the minimum separation for $\mathsf{Stab}_\epsilon$ is within a small factor of $\delta_G$ itself.

## A.2 Proof of Proposition 5.1

*Proof.* We have that $Z = (X, Y)$ is a data distribution in which by our assumption 2. preceding the proposition, we have that almost surely $Y = y^*(X)$ for a deterministic function $y^*$. With this notation, we may write

$$\mathbb{E}_Z[f(Z)] = \mathbb{E}_{X_\epsilon^0}\mathbb{E}_{g|X_\epsilon^0}[f(g \cdot X_\epsilon^0, y^*(g \cdot X_\epsilon^0))].$$

Recalling that we restrict to functions of the form $f(x, y) = \ell(\widetilde{f}(x), y) = \mathsf{d}_{\mathcal{Y}}(\widetilde{f}(x), y)^2$, we first consider the precise equivariance case $\epsilon = 0$. In this case for $g \in \mathsf{Stab}_\epsilon(\mathcal{F})$ we find also $\widetilde{f}(g \cdot X_\epsilon^0) = g \cdot \widetilde{f}(X_\epsilon^0)$ and thus when optimizing over $\widetilde{f}$ we have to determine the optimal value of $y = \widetilde{f}(X_\epsilon^0)$ to be associate to each $X_\epsilon^0$. Thus as a consequence of all the above, if $\widetilde{\mathcal{F}}$ would be the class of all precisely $\mathsf{Stab}_\epsilon$-equivariant measurable functions, we would get the following rewriting:

$$\mathsf{AppGap}(\mathcal{F}, \mathcal{D}) = \min_{\widetilde{f} \in \widetilde{\mathcal{F}}} \mathbb{E}_Z[\mathsf{d}_{\mathcal{Y}}(\widetilde{f}(X), y^*(X)] = \mathbb{E}_{X_\epsilon^0} \min_{y \in \mathcal{Y}} \mathbb{E}_{g|X_\epsilon^0}\left[\mathsf{d}_{\mathcal{Y}}(g \cdot y, y^*(g \cdot X_\epsilon^0)^2\right]. \quad (A.6)$$

For $\epsilon > 0$, for each fixed $X_\epsilon^0 = x_\epsilon^0$ we may further perturb the associated $y = \widetilde{f}(x_\epsilon^0)$ by at most $\epsilon$ in the direction of $y^*(X_\epsilon^0)$, while still obtaining a measurable function with approximation $\ell_\infty$-norm error bounded by $\epsilon$, thus the above bound is improved to

$$\mathsf{AppErr}(\mathcal{F}, \mathcal{D}) \leq \mathbb{E}_{X_\epsilon^0} \min_y \mathbb{E}_{g|X_\epsilon^0}\left[\left(\mathsf{d}_{\mathcal{Y}}(g \cdot y, y^*(g \cdot X_\epsilon^0)) - \epsilon\right)_+^2\right], \quad (A.7)$$

as desired. In case $\widetilde{\mathcal{F}}$ contains a strict subset of measurable invariant functions with error $\epsilon$, we would only get an inequality instead of the first equality in (A.6) but we still have the same bound as in (A.7), and thus the proof is complete. □

## A.3 Proof of Theorem 5.2

*Proof.* We use the isodiametric inequality (5.1) in $G$, applying it to $\mathsf{Stab}_\epsilon(\mathcal{F})$ for $\mathcal{F} \in \mathcal{C}_{\epsilon,\lambda}$. Then by taking $\mathsf{Stab}_\epsilon(\mathcal{F}) = X$ which is optimal for inequality (5.1) we can saturate the two bounds (modulo discretization errors for discrete $G$) and we get

$$\frac{|\mathsf{Stab}_\epsilon(\mathcal{F})|}{|G|} \simeq \lambda, \quad \mathsf{diam}(\mathsf{Stab}_\epsilon(\mathcal{F})) \simeq C_G \lambda^{1/\mathsf{ddim}(G)}.$$

We next use Lipschitz deformation bounds and find that for all $x \in \mathcal{X}$ we have

$$\begin{aligned} \mathsf{diam}\left\{y^*(g \cdot x) : g \in \mathsf{Stab}_\epsilon(\mathcal{F})\right\} &\leq \mathsf{diam}(\mathsf{Stab}_\epsilon(\mathcal{F}))\, \mathsf{Lip}(y^*)\, L' \\ &\leq C_G \lambda^{1/\mathsf{ddim}(G)} \epsilon(\mathcal{F}))\, \mathsf{Lip}(y^*)\, L'. \end{aligned}$$

Then we use Proposition 5.1 for $\mathcal{F}$ and observe that when $g, X_\epsilon^0$ are random variables as in the proposition, in particular $g \in \mathsf{Stab}_\epsilon(\mathcal{F})$ and for each $X_\epsilon^0 = x_\epsilon^0$ we find the following estimate valid uniformly over $y \in \left\{y^*(g \cdot X_\epsilon^0) : g \in \mathsf{Stab}_\epsilon(\mathcal{F})\right\}$:

$$\mathsf{d}_{\mathcal{Y}}(y, y^*(g \cdot x_\epsilon^0)) \leq C_G' \lambda^{1/\mathsf{ddim}(G)}\, \mathsf{Lip}(y^*)\, L'.$$

In a similar way, we also find

$$\mathsf{d}_{\mathcal{Y}}(y, g \cdot y) \leq C_G' \lambda^{1/\mathsf{ddim}(G)}\, L'.$$

By triangle inequality, and using the assumption that $\mathsf{Lip}(y^*) \simeq 1$ it follows that

$$\mathsf{d}_{\mathcal{Y}}(y, y^*(g \cdot x_\epsilon^0)) \leq C_G' \lambda^{1/\mathsf{ddim}(G)}(1 + \mathsf{Lip}(y^*))L' \lesssim C_G' \lambda^{1/\mathsf{ddim}(G)}\, \mathsf{Lip}(y^*)\, L'.$$

Then we may perturb each $y$ by $\epsilon$ in order to possibly diminish this value without violating the condition defining $\mathsf{Stab}_\epsilon(\mathcal{F})$, and with these choices we obtain the claim. □

## A.4   Finding the optimal $\lambda = \lambda^*$ for the bound of Theorem 6.1

We note that $\lambda^*$ minimizing $C\lambda^\alpha + C'\lambda^{-\beta}$ for $\alpha, \beta > 0$ is given by

$$\lambda^* = \left( \frac{\beta}{\alpha} \frac{C'}{C} \right)^{1/(\alpha+\beta)} .$$

recall that in our case have the following choices, for case 1 and case 2 in the theorem's statement.

$$\alpha_1 = \alpha_2 = 1/d_G, \qquad \beta_1 = 1/2, \beta_2 = 1/d_0,$$

and

$$\begin{aligned}
C &= C_G \mathsf{Lip}(y^*) L', \\
C_1' &\simeq \frac{(2LD)^{d/2} |G|^{1/2}}{\delta_G^{(d_0-2)/2}}, \\
C_2' &\simeq (2LD)^{d/d_0} |G|^{1/d_0},
\end{aligned}$$

and thus the optimal choice of $\lambda$ is

$$\text{in case } n\lambda \geq C_3, \quad \lambda^* = \left( \frac{2}{d_G} \frac{(2LD)^{d/2} |G|^{1/2}}{\delta_G^{(d_0+2)/2}} \right)^{2d_G/(d_G+2)} n^{-d_G/(d_G+2)},$$

$$\text{in case } n\lambda > C_3, \quad \lambda^* = \left( \frac{d_0}{d_G} (2LD)^{d/d_0} |G|^{1/d_0} \right)^{d_0 d_G/(d_0+d_G)} n^{-d_G/(d_G+2)}.$$

## B   Examples

We describe some concrete examples of partial and approximate equivariance using the language we used in section 3.2 while sourcing them from existing literature. But first, we expand a little on our equivariance error notation.

### B.1   Equivariance error notation

Recall that the action of elements of an ambient group $G$ over the product space $\mathcal{Z} = \mathcal{X} \times \mathcal{Y}$ may be written as follows: For coordinates $z = x \times y$ we may write $g \cdot z = (g \cdot x, g \cdot y)$, and thus for $\widetilde{f} : \mathcal{X} \to \mathcal{Y}$ and $f(x,y) = \ell(\widetilde{f}(x), y)$, we have the action

$$(g \cdot f)(z) := f(g \cdot z) = \ell(\widetilde{f}(g \cdot x), g \cdot y).$$

For the equivariance error of $g, f$, interpreted as "the error of $f$'s approximate equivariance under the action by $g$", we get the following, which is valid in the common situations in which $\ell(y, y') \geq 0$ in general with $\ell(y, y) = 0$ for all $y \in \mathcal{Y}$:

$$\mathsf{ee}(f, g) := \|g \cdot f - f\|_\infty = \sup_{x,y} \left| \ell(\widetilde{f}(g \cdot x), g \cdot y) - \ell(\widetilde{f}(x), y) \right| \geq \sup_x \ell(\widetilde{f}(g \cdot x), g \cdot \widetilde{f}(x)), \text{ (B.1)}$$

where the last inequality follows by restricting the supremum from $\mathcal{X} \times \mathcal{Y}$ to the graph of $\widetilde{f}$, namely by imposing $y = \widetilde{f}(x)$.

In several recent works, the equivariance error is defined simply by comparing $g \cdot \widetilde{f}(x) and \widetilde{f}(g \cdot x)$, as in the rightmost term of (B.1), thus it is lower than the one found here. We provide a justification for our definition of the equivariance error:

- The loss $\ell$ is the integrative part of the model, thus a definition for equivariance error which does not include it will only detect partial information concerning the influence of symmetries.

- The notion of equivariance error defined via $\ell$ simplifies the comparison between $\widetilde{f}$ and data distributions $\mathcal{D}$.

### B.2    Examples

#### B.2.1    Imperfect translation equivariance in classical CNNs

We consider here the most common examples of group equivariant convolutional networks (GCNNs), which are the usual Convolutional Neural Networks (CNNs) for computer vision tasks. We follow observations from [27] and [11], and connect the underlying ideas to Theorem 6.1.

**Setting of the problem.**    We consider a usual CNN layer, keeping in mind a segmentation task, where both $\mathcal{X} = \mathcal{Y}$ represent spaces of images. More precisely, we think of images as pixel values at integer-coordinate positions taken from the index set $\mathbb{Z} \times \mathbb{Z}$. We also assume that the relevant information of each image only comes from a square of size $n \times n$ pixels, outside which the pixel values are taken to be 0. We consider the application of a single convolution kernel/filter, of $k \times k$ pixels (with $k$ a small odd number). One typically applies padding by a layer of 0's of size $(k-1)/2$ on the perimeter of the $n \times n$ square, after which convolution with the kernel is computed on the $n \times n$ central pixels of the resulting $(n+k-1) \times (n+k-1)$ padded input image. The output relevant information is restricted to a $n \times n$ square, outside which pixel values are set to 0 again, via padding.

**Metric on $\mathcal{X}$.**    As a natural choice of distance over $\mathcal{X}$ we may consider $L^2$-difference between pixel-value functions, or interpret pixel values as probability densities, and use Wasserstein distance, or consider other ad-hoc image metrics.

**Group action: translations.**    The group acting on our "pixel-value functions" is the group of translations with elements from $\mathbb{Z} \times \mathbb{Z}$. We expect the following invariance for the segmentation function $f : \mathcal{X} \to \mathcal{X}$:

$$f(v \cdot x) = v \cdot f(x),$$

where $x \in \mathcal{X}$ represents an image with pixel values assigned to integer coordinates and $v \in \mathbb{Z}^2$ is a translation vector and (in two alternative notations) $v \cdot x = \tau_v(x)$ is the result of translating all pixel values of $x$ by $v$.

**Deformation properties of the action.**    If we take the previously mentioned distance functions on $\mathcal{X}$ and the usual distance induced from $\mathbb{R}^2$ over translation vectors $v$, it is easy to verify that the assumptions of Theorem A.3 about the action of translations hold, and the Lipschitz constants with respect to the metric on $\mathcal{X}$ only depend on the mismatch near the boundary, due to "zero pixels moving in" and to "interior pixels moving out" of our $n \times n$ square, and being truncated. The ensuing bounds only depend on the precise distance that we introduce use on $\mathcal{X}$.

**More realistic actions.**    An alternative more realistic definition of $\mathbb{Z} \times \mathbb{Z}$-action consists of defining $v \cdot x$ as the truncation of $\tau_v(x)$ where, for pixels outside our "relevant" $n \times n$ square we set pixel value to 0 after the translation.

**Problems near the boundary.**    Nevertheless, the updated translation action, will move pixel values of 0, coming from pixels outside the $n \times n$ square, and will create artificial zero pixel values inside the image $v \cdot x$, different than the values that would be present in real data.

**Imperfect equivariance of data.**    Also, even in the latter more realistic alternative, the above translation equivariance is not respecting by segmentation input-output pairs coming from finite $n \times n$ images, since, independently of $n$, the boundary pixel positions translated by $v$, fall outside the original image.

**Approximate stabilizer.**    In any case, we have to restrict the choices of $v$ to integer-coordinate elements of a (subset of a) $n \times n$ square, containing only the translations that are relating "real" segmentations that appear within our $n \times n$ relevant window. It is thus

natural to restrict $\mathsf{Stab}_\epsilon$ to only include vectors in a smaller subset of $\mathbb{Z} \times \mathbb{Z}$, of cardinality

$$|\mathsf{Stab}_\epsilon| \leq n^2.$$

The value of error $\epsilon$ may quantify the allowed error (or noisiness) for our model sets.

**Reducing to finite $G$ and computing $\lambda$.** For the sake of computing $\lambda$ in our Theorem 6.1, we can observe that input and output values outside the "padding perimeter" given by a $(n + k - 1) \times (n + k - 1)$ square, are irrelevant, and thus we may actually periodize the images and consider the images as subsets of a *padded torus* $(\mathbb{Z}/(n + k - 1)\mathbb{Z})^2$, which we consider as acting on itself by translations. In this case $G \simeq (\mathbb{Z}/(n + k - 1)\mathbb{Z})^2$, so that

$$|G| = (n + k - 1)^2 \quad \text{and thus} \quad \lambda = \frac{|\mathsf{Stab}_\epsilon|}{(n + k - 1)^2}.$$

**Further extensions.** In [27], it is argued that convolutional layers in classical CNNs are not fully translation-equivariant, and can encode positional information, due to the manner in which boundary padding is implemented. Possible solutions are increasing the padding to size $k$ (so that the padded image is a square of size $(n + 2k) \times (n + 2k)$) or extending images by periodicity, via so-called "circular padding" (which transforms each image into a space equivalent to the torus $(\mathbb{Z}/n\mathbb{Z})^2$). In either case, the application of actions by translations by vectors that are too large compared to the image size of $n \times n$, will increase the mismatch between model equivariance and data equivariance.

**Stride and downsampling.** In [11], a different equivariance error for classical CNNs is studied, related to the use of stride $> 1$ in order to lower the output dimensions of CNN layer outputs. If for example we use stride 2 when defining layer operation $f : \mathcal{X} \to \mathcal{Y}$, then $\mathcal{Y}$ will have relevant pixel values only in an $n/2 \times n/2$-square, and we apply the $k \times k$ convolution kernel only at positions with coordinates in $(2\mathbb{Z}) \times (2\mathbb{Z})$ from image $x$. In this case we require that translations by group elements $v \in (2\mathbb{Z}) \times (2\mathbb{Z})$ on $x$ have the effect of a translation by $v/2 \in \mathbb{Z}$ on the output. However for shifts in the input via vectors $v$ that do not have two even coordinates, we may not have have an explicit corresponding action on the output, and in [11] a solution via adaptive polyphase sampling is proposed. A possibility for studying the best polyphase sampling strategy via almost equivariance, would be to include a bound for equivariance error $\epsilon > 0$ and consider the optimization problem of finding the polyphase approximation that minimizes theoretical or empirical quantifications of $\epsilon$. As a benchmark (modelled on the case of infinite images without boundary effects) one could compare the above to the action via $\mathsf{Stab}_0 = (2\mathbb{Z}) \times (2\mathbb{Z})$ which has $\lambda = 1/4$ within the ambient group $G = \mathbb{Z} \times \mathbb{Z}$. Our Theorem 6.1 can be used to compare the effects of increasing or decreasing $\epsilon, \lambda$, in terms of data symmetry.

### B.2.2 Partial equivariance in GCNNs

In this section, we connect the main results from [40] to our setup. In [40], one of the main motivating examples was to consider rotations applied to a handwritten digit and revert them. The underlying group action was via $SO(2)$ and only rotations of angles between $[-60°, 60°]$ were permitted in one case, which allowed to not confound rotated digits "3" and "9" for example.

The above task can be formulated on a space of functions $f : \mathcal{X} \to \mathcal{Y}$ in which $\mathcal{X}$ represents the space of possible images and $\mathcal{Y}$ the labels. Elements $(Y_d, Y_\theta) \in \mathcal{Y}$ include a digit classification label $Y_d$ and a rotation angle value $Y_\theta$.

We consider actions by group $G = SO(2) = \{R_\phi : \phi \in \mathbb{R}/360\mathbb{Z}\}$, where $R_\phi$ is the rotation matrix by angle $\phi$, and the group operation corresponds to summing the angles, modulo $360°$ (or in radians, modulo $2\pi$). The action of $R_\phi$ over $\mathcal{X}$ would be by rotation as usual ($R_\theta$ sends image $x \in \mathcal{X}$ to $R_\theta \cdot x$, now rotated by $\theta$), and over $\mathcal{Y}$ we consider the action by

$$R_\phi(Y_d, Y_\theta) = (Y_d, Y_\theta + \phi),$$

i.e. the restriction of the action on the digit label leaves it invariant and the restriction of the action on the angle label is non-trivial, giving a shift on the label.

The optimum labelling assigns to $x$ a label $y^*(x)$ enjoying precise equivariance under the above definitions of the actions, and thus we are allowed to permit equivariance error $\epsilon = 0$. However as mentioned above, applying rotations outside the range $\theta \in [-60°, 60°]$ to the data would surely bring us outside the labelled data distribution, thus we are led to take

$$\epsilon = 0, \quad \mathsf{Stab}_0 = \{R_\theta : \ \theta \in [-60°, 60°]\}.$$

We then have that $|\mathsf{Stab}_0|/|SO(2)| = 1/3$, with respect to the natural Haar measure on rotations. It is natural to think of the set of $\mathsf{Stab}_0$-action representatives of images $\mathcal{X}_0^0$ given as the "unrotated" images. If we take a digit image that is rotated, say by $20°$, from its "base" version, and we apply a rotation of $50°$ to it (i.e. an element of $\mathsf{Stab}_0$), then we reach the version of the image now excessively rotated by $70°$. This means that without further modification, considering model symmetries with $\mathsf{Stab}_0$ taken to be independent of the point, would automatically generate some error when tested on the data. While decreasing the threshold angle in the definition of $\mathsf{Stab}_0$ from $60°$ would limit this effect, it will also decrease generalization error in the model. The study of point-dependent invariance sets $\mathsf{Stab}_\epsilon$ is interesting in view of this example application, but it is outside the scope of the current approximation/generalization bounds and is left for future work.

### B.2.3 Possible applications to partial equivariance in Reinforcement Learning

The use of approximate invariances for RL applications was considered in [23, Sec. 6] via soft equivariance constraints allowing better generalization for an agent moving in a geometric environment. While imposing approximate equivariance for memoryless $G$-action for groups such as $G = SO(2), \mathbb{Z}_2$ has produced positive results, it may be interesting, in analogy to the previous section, to include memory, and thus restrict the choices of group actions across time steps. Note that for a temporal evolution of $T$ steps, the group action by $G$ acting independently at each step would produce a $T$-interval action via the product group $G^T$, and allowing for a partial action via $\mathsf{Stab}_\epsilon$, with possibly increased fitness to evolving data. More precise time-dependence prescriptions and consequences within $Q$-learnig are left to future work.

## C  Discussion of [20]

In section 2 we mentioned [20], which considers PAC-style bounds under model symmetry. [20] works with compact groups and argues how the learning problem in such a setup reduces to only working with a set of reduced orbit representatives, which leads to a generalization gain. This message of [20] is similar to ours, although we work with a more general setup. However, we noted that the main theorem in [20] has an error. Here, we briefly sketch the issue with the proof.

One of the main quantities of interest in the main theorem of [20] is $D_\tau(\mathcal{X}, \mathcal{H})$ (notation from their paper), which directly comes from the following bound, and is claimed to have a linear dependence on $Cov(\mathcal{X}, \rho, \delta)$. Again, for the sake of easier verification, we follow their notation. Crucially, note that [20] uses the notation $Cov$ as analogous to our $\mathcal{N}$:

$$Cov(\mathcal{H}, \|\cdot\|_{L_\infty}, 2C\delta + \kappa) \leq Cov(\mathcal{X}, \rho, \delta) \sup_{x \in \mathcal{X}} Cov(\mathcal{H}(x), \|\cdot\|_\infty, \kappa)$$

However, the correct application of the Kolmogorov-Tikhomirov estimate shows that the reasoning in the proof should yield a dependence which is exponential in $Cov(\mathcal{X}, \rho, \delta)$, not linear. To see this, set $s = 2$ (sufficient for our purposes) in equation 238 in [54] (page 186). In other words, it is not possible to cover Lipschitz functions in infinity norm by only using constant functions.

