# OpenReview forum: "Approximation-Generalization Trade-offs under (Approximate) Group Equivariance"
_NeurIPS.cc/2023/Conference — NeurIPS 2023 poster_

### Official Review · Reviewer_PxmR · 2023-07-02

**Soundness:** 4 excellent
**Presentation:** 3 good
**Contribution:** 4 excellent
**Rating:** 8
**Confidence:** 2

**Summary:**

The paper presents a theoretical investigation into the approximation and generalization errors for a model exhibiting approximate equivariance. The paper introduces the concept of "error to equivariance," a measure of the deviation of the error function from perfect equivariance. It then defines the $\epsilon$-stabilizer, a subset of a group under whose transformations the error function maintains approximate equivariance. Leveraging these definitions, the paper dives into the analysis of generalization error for an approximately equivariant model in terms of PAC-bounds and Rademacher complexity, after which, It also explores the approximation error. Ultimately, the performance error of a model is defined as the sum of its generalization error and approximation error, providing a quantitative measure of the performance of an approximate equivariant model. In the end, the performance error can quantitatively analyze the trade-off between its two terms (generalization error and approximation error).

**Strengths:**

1. The paper theoretically studies approximate equivariance, which is a very significant problem in the equivariant learning literature as real-world problems are not always perfectly equivariant.
2. The paper provides general bounds, contributing valuable theoretical tools to the field.

**Weaknesses:**

1. Section 4 is somewhat challenging to read, especially for readers who are not familiar with PAC-bounds and Rademacher complexity. Additional explanations or intuitive examples could improve accessibility.
2. While not a requirement for a theory paper, the absence of empirical validation or experiments leaves a gap. Experiments that align with the theoretical bounds could provide practical guidance on how the paper's findings might be applied empirically.

**Questions:**

1. Could the authors comment on the broader implications of the proposed theory? Specifically, could the theory help model selection under approximate equivariance?
2. The paper is titled "Approximation-Generalization Trade-offs", however, the high-level message regarding this trade-off isn't clear.
3. In line 140, there is a closing parenthesis but no opening parenthesis.

**Limitations:**

The paper properly discusses its limitations. An additional limitation to consider is the assumptions of the paper. The paper makes a couple of assumptions throughout the theoretical analysis, it might be good to summarize and discuss those assumptions in the limitation section.

---

> ### Author Rebuttal · Authors · 2023-08-09
>
> We thank the reviewer for the very encouraging feedback and recognizing the importance of our work. We try to address each of the points raised one by one below:
>
> ### Section 4
> Thanks for pointing out that accessibility could be improved! Since section 4 mostly describes background, we decided to compress it a little bit due to space constraints and get more space for newer results. To address this, we can add a refresher section in the appendix. If the reviewer has other instructions, we would be glad to consider them for incorporation.
>
> ### Empirical Validation.
> As we also mentioned to reviewers CLQc and 5syP, we thought seriously about including some experiments. However, we eventually took a call to not include experiments due to a couple of main reasons. First, the results that we present are really the first in formally presenting a common perspective in equivariant learning -- which is often stated in introductory sections -- but hasn't quite been formalized. We provide results for both generalization and approximation errors for the general case when the symmetries need not form a group, and then use them to work out a tradeoff which quantifies the effect of model misspecification. In fact, even if we consider only generalization bounds, existing results are limited to exact equivariance, and even there deal with special cases. Given that we go a few steps further for generalization, and then add results on approximation as well, we thought our results might be sufficient enough for a standalone contribution. Second, a flipside of the generality of our results is that they don't propose a specific prescription for equivariance/partial equivariance/approximate equivariance. If we provide experiments, they would amount to fixing a specific scheme for partial equivariance, so we eventually decided against it. However, to support our main result, we could consider some of the concrete cases we have described in the appendix e.g. in Finzi et al [1], cast it in terms of our main results (which we had explored while writing the paper) and run some modified versions of the experiments they use in their paper. If accepted, we will consider adding an experiment such as this in the appendix and refer to it right after theorem 6.1.
>
> ### Model selection
> We will add some comments on this. Since we do a PAC-style analysis, and the models we treat are quite general, the guidance is somewhat weak. However, if we fix stronger structural hypothesis to obtain the equivalent of theorem 6.1 in specific settings, we could get more tangible directions for model selection. This is an avenue that we are currently exploring. We will add some comments on this in "limitations", if accepted.
>
> ### Tradeoffs
> We will be sure to be more clear about the high-level message about this tradeoff i.e. the effect of model misspecification both in the beginning and end of the paper. However, just to reiterate: In section 4 of the paper we basically show that if we allow a reasonably rich set of approximate symmetries of the model, we get better generalization. Note that this part holds for any data distribution and makes no assumptions about the symmetries of the data. In section 5, we look at the approximation error. If we wish to have small approximation error, we need to permit more freedom on the symmetries. We measure this by the parameter $\lambda$ which bounds the size of the stabilizer $\mathsf{Stab}_\epsilon$ relative to the full group $G$. The approximation error also includes a dependence on the error in our symmetries, measured by $\epsilon$. However, even here, we we still do not fix any explicit symmetry properties for the data. This implies that we can guarantee that there exists at least one efficient model efficient model such that its symmetries will necessarily fit to those of the data. Thus, our theorem 5.2 implies that independent of data symmetries, the improvement that we see due to "symmetry alignment" is bounded. When we combine both results in theorem 6.1, this tradeoff becomes more explicit. We can definitely add more comments to make it more explicit. Let us know if you would have specific guidance on what you would like to see with regards to your question.
>
> ### Typos
> Thanks for pointing these out -- we have fixed them.
>
> ### Limitations
> Thanks for the suggestion! We will also list our assumptions in the limitations section. Some of these might also be more interesting for future work. For instance, our bounds work well for groups that are nicely doubling e.g. locally compact groups, but making them work for permutation groups requires more careful analysis since their metric structure is not easy to treat in the same manner as we use in our paper.
>
>
> References cited in this comment:
>
> References: [1] Marc Finzi, Gregory Benton, and Andrew G Wilson. Residual pathway priors for soft equivariance constraints. In M. Ranzato, A. Beygelzimer, Y. Dauphin, P.S. Liang, and J. Wortman Vaughan, editors, Advances in Neural Information Processing Systems, volume 34, pages 30037–30049.

---

> > ### Comment · Reviewer_PxmR · 2023-08-11
> >
> > Thank you for the rebuttal. Most of my concerns are resolved, and I will keep my recommendation as strong accept.

---

> > > ### Author Response · Authors · 2023-08-12
> > > **Thanks!**
> > >
> > > Dear Reviewer,
> > >
> > > Thanks for taking the time to read our rebuttal and acknowledging it! We are thankful for your feedback. Please let us know if we can provide any further clarifications or if you have more suggestions vis-à-vis the presentation of our results.

---

### Official Review · Reviewer_5syP · 2023-07-04

**Soundness:** 3 good
**Presentation:** 3 good
**Contribution:** 3 good
**Rating:** 7
**Confidence:** 3

**Summary:**

In this paper, the authors spell out explicitly the benefit of capturing task specific symmetries into machine learning models. This goal is achieved by deriving rigorously  generalization error bounds that takes into account the cardinality of the set of transformations that a model is (approxaimately) equivariant to (Stab_\epsilon) as well as the group dimensionality these transformations are elements in.
The results are very general and unlike previously presented results in the literature, the sets of transformation the model is equivariant to, are not required to form a group.

**Strengths:**

The presented approach to derive the bounds is principled and very interesting.


**Weaknesses:**

While the derived results are very general and rigorous, it is not directly obvious how these results can be applied. Although the authors give examples in the appendix, the impact of these bounds seems relatively limited. It could have been maybe useful to show an actual model performance w.r.t. the presented bound.


**Questions:**

Is there anything that can be said about the tightness of the presented bounds?


**Limitations:**

The authors state that the derived bounds are not model specific and stronger versions could be the subject of future work.

---

> ### Author Rebuttal · Authors · 2023-08-09
>
> Thank you for your appreciative comments about our work and recognizing its contribution to the literature. We hope to address some of the questions raised point-by-point below.
>
> ### Applicability and experiments evaluation
> As we mentioned to reviewer CLQc, we have considered including experiments quite seriously. However, since this is the first paper to address the tradeoff between generalization and approximation in a general equivariant setting, we feel that its role is to establish intuition and establish the important principles at work. Note that existing works mostly only deal with generalization, only with precise equivariance, and in that restriction, only with special cases. However in most real-life cases, we do not have neither full nor precise symmetry: the value of our paper is to see how to fit this in theoretical estimates. Further, our results cover a full range of generality, we have the weakest available assumptions on the form of the action of the symmetry group on the involved spaces. In particular note that there is no classification of the types of actions that might occur. As mentioned in the response to reviewer CLQc, if required we may implement modified versions of the experiments of reference [1], and consider adding these experiments comparing them to Theorem 6.1.
>
> ### Q1 tightness of upper bounds
> Also see our response to reviewer W1Je. Indeed, as in standard PAC theory, we prove upper bounds independent on the data distribution. The insight in the proofs is that we rigorously illustrate the interplay/tradeoff between approximation and generalization bounds in the setting of equivariance. To prove tightness of the bounds we would have to build problematic data distributions, and show that the PAC bounds worst case scenario is sharp. This is not done in PAC theory, and we did not pursue it here, the motivations being that the so-constructed datasets would require very technical, abstract and lengthy constructions and would likely be unrealistic. Thus they would shed very limited (if any) light on phenomena that can be useful for applications. We are currently exploring stronger results that work for specific groups and under stronger structural hypothesis. But such results will still be upper bounds, but just sharper for specific use cases. The bounds can also be improved in other ways, such as being made data-dependent.
>
> References used in this comment:
>
> [1] Marc Finzi, Gregory Benton, and Andrew G Wilson. Residual pathway priors for soft equivariance constraints. In M. Ranzato, A. Beygelzimer, Y. Dauphin, P.S. Liang, and J. Wortman Vaughan, editors, Advances in Neural Information Processing Systems, volume 34, pages 30037–30049.

---

> > ### Comment · Reviewer_5syP · 2023-08-17
> >
> > I appreciate the effort put in the rebuttal and the clarifications. I do see the value of the presented bound and appreciate the mentioned challenges in deriving tightness results which is something that you might consider adding as limitation in the introduction or concluding remarks or explicit limitations/assumption paragraph or section as suggested by CLQc. Implementing a modified versions of the experiments of reference [1] would be great but I wouldn't call it "required". My initial rating was that the paper is technically solid and I would keep the same rating.

---

> > > ### Author Response · Authors · 2023-08-18
> > > **Response**
> > >
> > > Dear Reviewer,
> > >
> > > Thank you for taking the time to go over our rebuttal(s). We are glad that we were able to convey at least some of the value of the presented bounds. We would also like to note that an alternative to some of the directions we mentioned in the response could also possibly include something inspired by https://proceedings.neurips.cc/paper_files/paper/2022/file/5a829e299ebc1c1615ddb09e98fb6ce8-Paper-Conference.pdf this is however, currently not on our radar. Having said that, we will certainly add what you mention to the limitations, which we are planning to add as a separate explicit subsection. Lastly, for the experiments, we have been planning some, and thinking how to fit them in the narrative of theorem 6.1. If we add experiments to the appendix based on [1], we will point to them below theorem 6.1. Although we do think that pursuing empirical work would be more appropriate for stronger results (and as we mentioned in one of the rebuttals, we are pursuing this direction in an NLP context).

---

### Official Review · Reviewer_R2KT · 2023-07-24

**Soundness:** 2 fair
**Presentation:** 3 good
**Contribution:** 3 good
**Rating:** 6
**Confidence:** 3

**Summary:**

This paper provides theoretical investigation on the intuition that, machine learning models (e.g. neural networks) whose symmetry aligns with the data symmetry achieve best performance and generalization. The paper formulate the relationship between model equivariance error and data equivariance error by providing quantitative bounds, which are applicable in the presence of approximate or partial data/model equivariance.

**Strengths:**

1. To my best knowledge, the paper provides a systematic and comprehensive review of the historical development of neural networks with group symmetries (**since this is not exactly my area of research, please correct me if I'm wrong**).

2. The paper provides detailed and systematic analyses of the model generalization bounds as well as approximation bounds under the more general case of partial or approximate equivariance. According to the authors, as well as to my best knowledge, this is the first theoretical work that addresses this problem. Therefore its contribution and novelty is significant.

3. Although not being able to check every technical details and derivations, I find the paper well-structured and easy to follow.

**Weaknesses:**

**Major issues:**

1. Derivation of the approximation error bounds:

- First of all, in line 291, I don't think it's legal to write $\min A - \min B$ as $\min(A-B)$, since the second term has a minus sign. So I think the correct way to write is that

    $\text{AppGap}(\mathcal{F}, \mathcal{D}) := \underset{f\in\mathcal{F}}{\min}\mathbb{E}[\mathbb{E}_g[f(g\cdot Z)]] - \underset{F\in\mathcal{M'}}{\min}\mathbb{E}[F(Z)] \ge \underset{F\in\mathcal{M'}}{\min}\mathbb{E}[\mathbb{E}_g[F(g\cdot Z)]] - \underset{F\in\mathcal{M'}}{\min}\mathbb{E}[F(Z)] $

- The expression above provides a **lower bound** to the approximation gap, so how does the inequality become equality in Line 304? If it is derived by assuming $\mathcal{F} = (\mathcal{M}^{(G)})'$ (assumption 3 in line 301=303), then line 304 should be written as

    $\text{AppGap}((\mathcal{M}^{(G)})',\mathcal{D}) = \underset{F\in(\mathcal{M}^{(G)})'}{\min}\mathbb{E}_Z[\mathbb{E}_g[F(g\cdot Z)]]$

- Moreover, if the upperbound in Prop 5.1. is derived based on the equation above, I think it's necessary to prove that $(\mathcal{M}^{(G)})'\subseteq \mathcal{C}_{\epsilon,\lambda}$ , for any $\epsilon\ge 0, \lambda\in(0, 1]$, which I don't see in the proof.

2. Interpretation of the "generalization-approximation trade-off"/"optimal equivariance" is not clear:

    In line 360-362, it is claimed that the optimal $\lambda^*$ "validates the intuition stated at the onset that for the best model its symmetries must align with that of the data". However, based on my understanding, the symmetries of the model **only fully align with those of the data** when $\lambda^*=1$, given $\epsilon=0$. When $0<\lambda^*<1$, I think it means the model only **partially preserves the symmetries of the data**. Do you agree? If this is the case, how do you justify the intuition that "the integration of relevant symmetry results in enhanced generalization."?

3. Similar to reference 9 in the paper, I think one key point I would expect to see in this paper is proving that, models with aligned group symmetries of data exhibit higher sample efficiency/complexity, which can generalize to scenarios of partial/approximate symmetry. I see no discussion or proofs regarding this intuition in the paper.

  **I will be happy to raise my score given satisfactory explanation or revision.**

**Minor Issues:**

1. This paper targets specifically on the theory of machine learning models with partial/approximate group symmetries, which might be of interests only to a very specific subgroup of the whole machine learning community.

2. There are some presentation issues need to be improved:

    - All hyperlinks do not work when clicked. Some numberings are also confusing, e.g. what is Thm A.4 in line 242, 246, 268?

    - Some of the statements would benefit from more citations/examples, e.g.
     "It has often been noted that this may impose an overly stringent constraint as real-world data seldom exhibits perfect symmetry” (line 36 - 37)
3. Typos

    - may have not -> may not have (line 312)

    - $\text{Stab}{\epsilon}(\mathcal{F'}) \Longrightarrow \text{Stab}{\epsilon}(\mathcal{F})$ (line 309)

**Questions:**

1. More regarding the interpretation of the "generalization-approximation trade-off"/"optimal equivariance":

    If we examine the contribution of $\lambda$ to the generalization and approximation bounds given by Theorem 4.4 and 5.2 respectively, assuming perfect symmetry of data ($\epsilon=0$), we see that the former decreases when $\lambda$ increases ($\frac{1}{\lambda^{1/2}}$ or $\frac{1}{\lambda^{1/d_0}}$ scaling), while the latter increases when when $\lambda$ increases ($\lambda^{1/d_G}$ scaling), which gives what the authors called "generalization-approximation trade-off". To me, this seems to show that minimizing the generalization error effectively aligns the model and data symmetries, whereas minimizing the approximation error functions in the opposite direction, resulting in a partially aligned model symmetry.

    Do you agree with this analysis? Any further explanation/intuition?

2. Can you explain what you mean by "oscillations", which is mentioned repeatedly? Example contexts like:

- "since $\text{Stab}_{\epsilon}$ includes more elements of G as ε increases, and we can expect that actions of elements generating higher equivariance error in general have higher oscillations". (line 274-275)

- line 320-322

---

> ### Author Rebuttal · Authors · 2023-08-09
>
> Thank you for your very encouraging comments on our work. We are grateful that you appreciate its significance and novelty and that you also liked our treatment/survey of the historical results in the area. :) We also found your questions instructive, and acknowledge their role in improving our presentation. We address each of the points raised below.
>
> ### Derivation of approximation error bounds
> This is a good catch. This was an oversight while presenting our results, and you are right that we have a wrong step in our derivation, as $ \min{A} - \min{B} \geq \min(A-B)$ and equality is not true in general. Thank you for pointing this out. Here is our response to this. For the replacement in our proofs, as mentioned in your report we can start with:
>
> $ \text{AppGap}(\mathcal{F}, \mathcal{D}) \ge \underset{F\in\mathcal{M'}}{\min}\mathbb{E}[\mathbb{E}_g[F(g \cdot Z)]] - \underset{F\in\mathcal{M'}}{\min} \mathbb{E}[F(Z)] (1). $
>
> From this point onwards, we can select $F^*$ which realizes the first minimum, and then we have:
> $\mathbb{E}[\mathbb{E}_g[F^*(g\cdot Z)]] = \underset{F\in\mathcal{M'}}{\min}\mathbb{E}[\mathbb{E}_g[F(g\cdot Z)]]$ and $  - \underset{F\in\mathcal{M'}}{\min} \mathbb{E}[F(Z)] \ge - \mathbb{E}[F^*(Z)].$
>
> Summing last two inequalities: $ \underset{F\in\mathcal{M'}}{\min}\mathbb{E}[\mathbb{E}_g[F(g\cdot Z)]] - \underset{F\in\mathcal{M'}}{\min} \mathbb{E}[F(Z)] \ge \mathbb{E}[\mathbb{E}_g[F^*(g\cdot Z)]] - \mathbb{E}[F^*(Z)].$
>
> Now the term on the right in the last inequality is a competitor for the minimization $\underset{F\in\mathcal{M'}}{\min}\mathbb E[\mathbb{E}_g[F(g\cdot Z)] - F(Z)],$ and we find $(1)\geq \underset{F\in\mathcal{M'}}{\min}\mathbb{E}[\mathbb{E}_g[F(g\cdot Z)] - F(Z)], $ as appears in our paper.
>
> ### inequality/equality
> What the referee writes is true, but line 304 in the paper is also correct. Note that we also have that $F\in(\mathcal M^{(G)})'$ is equivalent to $F(Z)=\mathbb E_g[F(g\cdot Z)]$ for all $Z$. Consequently, symmetrization map $F\mapsto [Z\mapsto\mathbb E_g[F(g\cdot Z)]$ is an involution from $\mathcal M'$ onto $(\mathcal M^{(G)})'$ and in particular all measurable invariant functions are all obtained by symmetrization:
> $(\mathcal M^{(G)})'=\{Z\mapsto\mathbb E_g[F(g\cdot Z)]:\ F\in \mathcal M'\}.$
> Thus, starting with the formula written by the referee, we also get
> $\min_{F\in (\mathcal M^{(G)})'}\mathbb E_Z[\mathbb E_g[F(g\cdot Z)]] = \min_{F\in \mathcal M'}\mathbb E_Z[\mathbb E_g[F(g\cdot Z)]]=\mathsf{AppErr}(\mathcal F,\mathcal D),$
> which is the formula claimed in the paper.
>
> ### upper bound/5.1:
> Assuming the above response to point 2 is satisfactory, the point raised by the referee after that "if the upper bound in Prop 5.1 is derived based on the equation above" (meaning, "based on the equation of point 2") should be automatically be resolved.
>
> ### optimal $\lambda^*$
> Indeed, in our formulation $\epsilon$ controls the allowed error in equivariance and $\lambda$ controls the proportion of elements of the group which are included in the computation of this error, thus measures the amount of symmetry included in the models. If we set $\epsilon=0$ then the minimum $\lambda^*$ for the error bound estimate from our theorem 6.1, indicates a value for the optimal proportion of group elements to be included in $\mathsf{Stab}_\epsilon$, i.e. for the measure of allowed symmetries in the models. If we take $\lambda>\lambda^*$ or $\lambda< \lambda^*$ this will worsen the error bounds, indicating that there exists a "precise quantity of symmetries" that one needs to include in the model for optimality: adding symmetries which would increase error because they are not present in the data, and adding too few symmetries will worsen generalization improvement and lead to higher error.
>
> ### explicit symmetry
> The reason for not including such explicit symmetry comparison in our proofs is that we focus on a higher generality than the one of reference [9] indicated above: we do not make strong assumptions on the groups or of the type of symmetries of the data. Thus we prove our bounds with minimal assumptions. An explicit comparison of model versus data symmetries would have increased the length and changed the focus of the paper, without deep changes in the fundamental concepts. We can include the following discussion in the additional space allowed for the final version of the paper, if accepted.
>
> For the precise comparison, we start by recalling that, roughly speaking, in Sections 4 and 5 respectively, we prove that
>
> (Sec. 4) If we allow a rich enough set of approximate symmetries of the model $\mathsf{Stab}_\epsilon$, we get better generalization bounds. Note that this part is valid for any data distribution, it does not require assumptions on symmetries of the data so far.
>
> (Sec. 5) If we wish to have small approx error, we need to allow freedom on the amount of symmetries. This is measured by $\lambda$ which bounds the size of the stabilizer $\mathsf{Stab}_\epsilon$$ relative to the full group $G$. The approx error also includes a dependence on the error in our symmetries, measured by $\epsilon$. Note that for the approx error bound we still do not fix any explicit symmetry properties for the data, therefore rather than proving a result valid for all models in class $\mathcal{C}_{\epsilon,\lambda} $, we guarantee the existence of at least one efficient model in this class. The underlying idea is that for an optimal model its symmetries will necessarily fit to those of the data (since we guarantee that this model class will approximate well the data). Therefore, our approximation bounds imply that, independently of the data symmetries, the improvement due to "symmetry alignment" is bounded as in Theorem 5.2.
>
> In future work it will be interesting to fix a specific class of symmetry groups; fix a quantification for the amount of symmetries present in data; make the approximation error dependence explicit, therefore making Theorem 5.2 more explicit.

---

> > ### Comment · Reviewer_R2KT · 2023-08-10
> > **Reply**
> >
> > I would like to thank the authors for providing satisfactory rebuttal comments. I think this work has significant contribution to the ML community and would like to recommend acceptance. However, more insight on the main result (especially the "generalization-approximation trade-off"/"optimal equivariance") and experimental evidence would make the paper stronger.

---

> > > ### Author Response · Authors · 2023-08-10
> > > **Thank you for the quick ack**
> > >
> > > Dear Reviewer,
> > >
> > > Thanks for again taking the time to look at our rebuttal. We are glad that we were able to address some of the question that you had raised.
> > > - We will certainly add more discussion around the results, discuss limitations, and add a sub-section explicating further on the "trade-off"/"optimal equivariance." If accepted, we can use the extra page to include these additions.
> > > - For the experiments, we will think how to structure them using the paper of Finzi et al. (as we have mentioned in a few comments here) and if accepted, we can include these in the appendix in the camera ready. As a side note: We were primarily thinking of scenarios where the interpretation is somewhat interesting/has not been studied before -- for this we identified a particular problem in NLP that we think is a great illustration of the benefit accorded by partial equivariance -- but were thinking of it in terms of future/ongoing work.

---

### Official Review · Reviewer_y3sQ · 2023-07-25

**Soundness:** 4 excellent
**Presentation:** 2 fair
**Contribution:** 3 good
**Rating:** 7
**Confidence:** 4

**Summary:**

In their theoretic work, the  authors presents quantitative bounds on the generalization error and approximation error. They extend the results to approximate symmetries, which refers to functions that are close, in the supremum norm, to a group action.

Their first results shows that when working with only the representatives of $G$-Orbits, and the models are equivariant (or almost-equivariant) The generalization error can be nearly-recovered.  Their second results show that their exists a class almost-equivaraint estimators, such that its approximation error can be bounded (and the bound depends mostly on the properties of G, and somewhat on the pertubations of the optimal estimator).


**Strengths:**

This paper is a novel and well presented work on approximate symmetry. The bounds presented are a significant contribution to the field, further deepening its theoretical foundations.


**Weaknesses:**

- The discussion section is lacking. It focuses on combining previous results and quantifying the lambda parameters. It is lacking interpretation of the results, and addressing your hypothesis. Under which conditions equivariance error is optimal? What is the interplay between data equivaraince to model equivariance? How does it compare to other bounds in the literature?
- The limitation of this paper is not addressed properly. The authors only mentions that "the results are not model specific", which is not an actual limitation. It would be beneficial to discuss the actual limitation of the work. Can the bounds be computationally intractable? for example, how can one compute $ddim(G)$? What is the asymptotic growth of the presented bounds?

- The paper is lacking intuition into the $\lambda$ parameter. Explanations are postponed into the discussion, and  even there, the interplay between $\lambda$      and the error is drawn, but there is no attempt to demystify this parameter. I would appreciate further elaboration about it, and about the meaning of $Dens(\epsilon)$. Is this the same as 'approximate orbits'$^{-1}$?


**Questions:**

- Theorem 5.2 holds under the condition that there \textbf{exists} $\mathcal{F} \in \mathcal{C}_{\varepsilon, \lambda}$ such that the theorem holds. I am concerned that ideally, we would like it to apply to \textbf{any} such $\mathcal{F}$ . It would seem to me that it would align better with our original motivation as we would like to prove that any such partial symmetry family (with limited perturbation) can generalize well. I note that theorem 6.1 is under the same assumption so my question additionally applies to it
-  I would appreciate further elaboration specifically on the interplay between model equavarince against data equavarince. How does the presented bound connect between them? can you show how further restriction of the models, choosing $\mathcal{F}' \subset \mathcal{F}$ would affect the generalization and approximation bound.


**Limitations:**

.

---

> ### Author Rebuttal · Authors · 2023-08-10
>
> We are grateful to the reviewer for a careful and critical reading of our paper and also for recognizing its significance in the literature on equivariance.
>
> ### Discussion
> We acknowledge that our presentation might appear a bit terse. As we mentioned to reviewer CLQc and W1Je, if accepted, we will use the additional page to add more context and discussion around each presented result. Since we were pressed for space, we took the following approach to presenting our results. We first spent the first two pages to give intuition about the problem and cover prior work in simplified terms. Then we review some basic PAC concepts and proceed to present our results step by step. We definitely agree that additional commentary could help the reader. If accepted, the additional page will help us do this, we can remove some parts and add more commentary before and after each result, and also discuss limitations in more detail. If you peruse our responses to other reviewers we have added more detail for additions.
>
> ### Comparison to bounds in the literature
> In the introduction and the related works section, we have tried to cover papers that present similar bounds and also briefly describe how they differ. After stating proposition 4.2, we point out below it, that it already generalizes existing bounds. The results after that are not directly comparable to existing work.
>
> ### tractability and ddim
> Could the reviewer kindly clarify, "computational tractability" of what is being referred to? We would like the opportunity to clarify further. About $\mathsf{ddim}(G)$, in the discussion around line 239 we briefly pointed out that for finite-dimensional Lie groups the doubling dimension corresponds to the usual dimension, which is known (these $G$ include rotation groups, Poincare groups, Euclidean isometry groups used in practice, or products thereof, amongst others). For more general $G$ the (polynomial/exponential) growth of groups is one of the main directions at the core of quite some deep work in Geometric Group Theory, which includes many answers in specific cases. However, the value for ddim will be known in most situations. We will be happy to add a remark about this in the extra space available, if accepted.
>
> ### Asymptotic Bounds
> Could you clarify what limit are you thinking about when you talk about asymptotic growth? If clear to us, we are happy to include a discussion.
>
> ### about results being model specific
> We here expand on the ideas of why not being model specific is a limitation for our results. As mentioned in the response to referee R2KT, if we would include more model-specific discussions, this would allow more in-depth control of what can happen as we increase/decrease the amount of allowed symmetries, or the approximate symmetry threshold $\epsilon$. To do this we would need to fix a model and to be specific about the action on the data and labels. Thus we decided against including this direction, with the motivation that it would have obscured the main interest of the paper, however we consider it a limitation because the description is less explicit. As a more precise example of this limitation, which we did not include due to space constraints, we note that for some classes of groups and actions, such as (products of) permutation groups acting on semantically equivalent token classes in language processing tasks, the bounds that we give using doubling dimension of the groups can be improved, using the specific form of the symmetry group and action in that case. In fact this seems to be a very interesting avenue of research, which we are currently exploring.
>
> We will consider expanding on these points, if accepted.
>
> We additionally will discuss more limitations, starting with the assumptions. For instance, our results require that the group under consideration has certain doubling properties which hold for groups such as locally compact groups. To prove analogous results as in our paper, for permutation groups, would require different analysis. This is an avenue for future work as it would involve understanding and relating the metric properties of permutation groups to the kind of concentration results that we use.
>
> ### Lambda parameter
> Thank you for pointing this out. We will add to our discussion around $\lambda$ if the paper is accepted. This parameter determines how many of the allowed symmetries (i.e. what percentage of the group $G$) are actual symmetries of our model class. The main intuition is that if $\lambda$ becomes smaller, in terms of class of model classes $\mathcal C_{\epsilon,\lambda}$ this means that we are relaxing the symmetry requirement on the model classes $\mathcal F$, thus the models in such $\mathcal F$ become more free to adapt to the data distribution. This increase of $\lambda$ describes what is usually called "partial equivariance" in the literature. On the other hand $\epsilon$ has the role of allowed error in the allowed symmetries (allowing so-called "approximate equivariance" in the models).
>
> ### Q1
> Please see above part on "lambda parameter". Also see our response to reviewers R2KT (under "explicit symmetry") and PxmR (under "tradeoffs"), since we seem to have run out of space in our rebuttal.
>
> ### Q2
> Please see above part on "lambda parameter".
> Second part of second question: We would appreciate if the reviewer could clarify what kind of restrictions in passing from $\mathcal F$ to $\mathcal F'$ the reviewer has in mind. Depending on this, the answer can be quite involved. If the restrictions are of the type of restricting the approximate equivariance error or making "less partial" the partial equivariance constraint (by increasing $\lambda$) then the relation is direct from the discussion after Thm. 6.1, which however we plan to explain in the appendix, if the paper gets accepted, in order to make it more explicit. If the referee intended something else, then we would be grateful if they can specify the question in more detail.

---

> > ### Comment · Reviewer_y3sQ · 2023-08-22
> >
> > Thank you for the detailed response and for the promise to add more context and discussion. About the passing from $\mathcal{F}$ to $\mathcal{F}'$, I  did meant 'making "less partial" the partial equivariance constraint', and I am happy to hear you plain to explain more in the appendix. My questions have been addressed and I retain my positive opinion of the paper.

---

### Official Review · Reviewer_W1Je · 2023-07-26

**Soundness:** 2 fair
**Presentation:** 1 poor
**Contribution:** 1 poor
**Rating:** 4
**Confidence:** 2

**Summary:**

The submission discusses the role of task-specific symmetries in enhancing machine learning models. It presents a quantitative analysis substantiating the common intuition that task-specific symmetries lead to improved generalization. The authors delve into the interplay between model and data symmetry, particularly in the case of partial or approximate symmetries, establishing a defined link between model equivariance error and data equivariance error. They outline the conditions that lead to the optimal model equivariance error. However, the results are general and not model-specific, with the authors suggesting future research to formulate more specific versions of their theorem tailored to specific groups and neural network architectures.

**Strengths:**

The paper provides a rigorous, quantitative exploration of task-specific symmetries in machine learning models.

**Weaknesses:**

The presentation of the paper's findings could be clearer, particularly with regard to illustrating their novelty and applicability, and the frequent use of different notations without a consolidated reference adds unnecessary complexity and potential confusion for readers.

**Questions:**

The paper's main contributions could be made more explicit. As it stands, the authors appear to enumerate their findings without adequately illustrating their novelty, importance, or applicability. Furthermore, the usage of numerous different notations creates some confusion; a table of notation might ameliorate this issue. While the paper seems to address a fundamental problem in machine learning theory, its presentation could be improved, and the clarity of its contributions could be enhanced.

Questions:

1. The majority of results in the paper are upper bounds, but there is no discussion regarding their tightness. Could the authors elaborate on this aspect?

2. In Lines 351-352, the authors mention that "the above bounds are not informative." If the bounds lack informativeness, could the authors clarify the significance or utility of these results?

---

> ### Author Rebuttal · Authors · 2023-08-09
>
> We appreciate the time that you have taken to read our paper and for providing feedback. Below we will try to address some specific points that you have raised.
>
> ### presentation
> As we mentioned to reviewer CLQc, if accepted, we will use the additional page to add more context and discussion around the results. We have a set of results that don't really exist in the literature (except in special cases), and given the limited space, we had to affect our own trade-off in terms of presentation. We eventually decided to present our results taking the following approach: We have spent the first two pages to properly state the problem and prior work in intuitive terms. Then we have reviewed existing prior works and tried to state their contributions, we then specify what issues have not been considered in the literature. Then we provide some basic background for PAC style bounds, and then proceed to present some results step by step. We definitely agree that additional commentary could help the reader. If accepted, the additional page will help us do this, we can remove some parts and add more commentary before and after each result, and also discuss limitations in more detail.
>
> ### novelty and applicability
> To add to the point above. We would like to emphasize that results on generalization for equivariant models are quite limited. The first part of our work on generalization proves results that subsume many existing results even if we restrict our results to the strict equivariance case. However, our results can also work with partial/approximate equivariance, which is relevant in real-world applications, so they are quite more general. We then go further and also provide approximation error bounds and show how they can be combined together to work out a trade-off i.e. quantifying the effect of model misspecification -- when the model and data symmetries don't match. Let us know if we should improve the discussion around this. In the first 2.5 pages, we have tried to contextualize our work such that it is clear what its contributions are and how they are situated in the literature. At the end of the review of basic PAC bounds, we state 'we are now in a position to state our results', implying that all the results stated after that are new.
>
> ### different notations without a consolidated reference
> Thanks. We will add a consolidated reference table in the beginning of section 3.
>
> ### Only upper bounds
> Yes, we treat worst case results in the spirit of standard PAC style analysis. Stronger results that work for specific groups and under stronger structural hypothesis is an avenue that we are currently exploring. Although such results will still be upper bounds, but just sharper for specific use cases (groups, architectures).
>
> ### Bound not informative
> We believe that the reviewer might have misread this line. We will consider rewording it. In fact, we say that "Note that the above bounds are not informative as $\lambda \to 0$ for the continuous group case, which corresponds to the second option in Theorem 6.1. This can be sourced back to the form of Assumption [...]", what we mean is that the bounds in theorem 6.1 are not useful when $\lambda$ is approaching zero, since the value of the upper bound tends to $\infty$. We don't mean to imply that the bounds are not informative in general. Kindly let us know if we have misunderstood something.

---

> > ### Comment · Reviewer_W1Je · 2023-08-17
> >
> > Thanks for the answers. I still don't see any numerical/theoretical evidence/explanation for the tightness of the upper bounds. An upper bound is only useful when it is tight. Therefore, I still doubt the usefulness of these upper bounds.
> >
> > As the authors said, the bound is useless when $\lambda\to\infty$. Does this imply that the upper bound is pretty loose when $\lambda$ is very small? This also related to my concern that the upper bounds in this paper can be very loose and useless in general.

---

> > > ### Author Response · Authors · 2023-08-18
> > > **Response**
> > >
> > > Dear Reviewer,
> > >
> > > Thank you for going over our rebuttal and for taking the time to respond. We would like to take the chance to further clarify why tightness is difficult in such settings (and perhaps not even desirable). To do so, we will first try to emphasize our main contribution and motivation. As we mentioned in some responses, we conceived of the work here as really being the first to address the approximation and generalization tradeoffs in very general equivariant settings. We think of our work's main contribution as establishing intuition and fleshing out some of the important principles at work. As we have noted, existing works deal with generalization in very restricted settings (group --> exact equivariance --> more restricted settings there too etc). Since we can not expect that most real-life situations would be exactly equivariant, we wanted to understand how this fits when we try to work out theoretical estimates. We instead try to cover a full range of generality, and make very weak assumptions vis-à-vis equivariance.
> > >
> > > With the above motivation, we work with standard PAC theory, using which we prove upper bounds that are distribution independent. As we mentioned to another reviewer, proving tightness would involve building pathological data distributions. Using these then, we would be able to show that the bounds in the worst case are sharp. However, this is not the focus in PAC theory. Building such distributions would require quite technical, abstract, and complicated constructions, which would also be unrealistic. It is not clear if such constructions could shed light on this phenomenon in such a manner that it is useful for applications. Exploring stronger results for specific groups is currently our focus, however. Such results would be sharper. We could also consider making our results data-dependent, although we haven't staked out this direction properly yet. For such stronger results, we are also looking at empirically-driven work as well. However, such work, we believe is out of the scope of this current paper.
> > >
> > > We appreciate you taking the time.

---

### Official Review · Reviewer_CLQc · 2023-07-31

**Soundness:** 4 excellent
**Presentation:** 3 good
**Contribution:** 4 excellent
**Rating:** 7
**Confidence:** 4

**Summary:**

There are two main results in this paper. The first one (Theorem 4.4) obtains a PAC-generalization bound for a hypothesis space which is approximately equivariant to transformations by elements g in G. This bound shows that in addition to the classical complexity term that depends upon the size of the hypothesis space, there are two additional terms that arise, one due to the approximate invariance (where one restricts to only the hypotheses that are equivariant to G) and another due to partial equivariance (where one relaxes the error to equivariance constraint). The proof of this result is classical (e.g., it is close to the Baxter’s proofs in “A Model of Inductive Bias Learning“).

The second result (Theorem 5.2) is about studying the interplay between equivariance imposed in the model and that possessed in the data. The key idea here is to construct a set of hypothesis spaces that capture a large subset of the "symmetries" in the data G, depending upon a parameter lambda; larger the lambda, smaller the set of hypotheses. This allows an upper bound on the approximation error in terms of lambda and the dimension of G.

**Strengths:**

This is a thorough and pertinent paper that asks the question: how does the equivariance baked into the model relate to the inherent symmetries present in the data. This is a rare perspective. The proofs are somewhat straightforward but they are well-formulated and rigorous.

**Weaknesses:**

The writing in Section 5 can be improved a lot.

I thought that the discussion in Theorem 6.1 could be done in a different way. Specific values of lambda and epsilon entail a choice of hypothesis space. The constant M in Theorem 6.1 therefore is smaller if lambda is larger (?).

Minor
I suspect the \in S on line 251 is a typo.
tilde cal F on line 289 and 298 should be just cal F I think.
cal F’ on line 309 should be cal F.
There is clearly a typo on Line 310.
C_G should be defined before Line 327.
Theorem 6.1 should have n^{-1/2} for the first term.

**Questions:**

It would be useful to demonstrate a meaningful experiment (even if it is with synthetic data) where the three terms in Theorem 6.1 trade off against each other.

---

> ### Author Rebuttal · Authors · 2023-08-09
>
> We are grateful to the reviewer for the very encouraging feedback. In particular, we are glad that our theoretical perspective on equivariance is appreciated and recognized to not have been commonly addressed. Thanks for the Baxter reference, which we were not aware of.
>
> Below we will address some specific points raised in the review.
>
> ### Writing in Section 5
> Thank you for your constructive feedback. We acknowledge that our discussion is a bit compressed since we were pressed for space and we had a lot of material to cover. On acceptance, the additional page will provide us some more leeway to improve the discussion in general -- in the discussion section, around each result, and also for the limitations. For section 5, to improve the discussion, we will undertake the following actions.
>
> #### Actions
> - Provide some more intuition when we state the AppGap term and the three core assumptions.
> - Add some commentary on the model classes defined before Proposition 5.1. Cite some standard results for the isodiametric inequality.
> - Add a few lines below theorem 5.2 explaining its relevance.
>
> We have worked on these additions, however, it seems like we are unable to submit a revised version of the paper during the rebuttal. Please let us know if you would like to see anything else added.
>
> ### Discussion in Theorem 6.1
> Thanks! This is an interesting suggestion. We think the discussion will certainly benefit by making it more explicit in the sense that you mention. We will add a few lines doing so. As a side note, we didn't quite follow the comment about "constant M being smaller", since it is fixed. Could the reviewer kindly clarify?
>
> ### Typos
> Thanks for catching it. We have fixed it and will propagate it when we are able.
>
> ### Experiments
> Including some experiments supporting our main results is something that we considered quite seriously. Eventually, we decided against it for two main reasons: First, we thought that the results are really the first in formalizing and supporting a common intuition in equivariant learning -- a perspective that can be commonly found in introductory sections of equivariance papers , but without formal backing. We provide results on generalization and approximation errors for the general equivariant case and then combine them to tease out a trade-off. Existing works mostly only deal with generalization (not approximation, or the generalization and approximation tradeoff) and even for the exact group equivariance case, deal only with special cases. So we believe that our results are reasonable contributions by themselves.  Second, our results are very general and don't propose a specific prescription for equivariance/partial equivariance/approximate equivariance. If we provide experiments, they would amount to fixing a specific scheme for partial equivariance, so we eventually decided against it. However, to support our main result, we will consider some of the concrete cases we have described in the appendix e.g. in Finzi et al [1], cast it in terms of our main results and run some modified versions of the experiments they use in their paper. If accepted, we will consider adding an experiment such as this in the appendix and refer to it right after theorem 6.1.
>
> References mentioned in this response:
>
> [1] Marc Finzi, Gregory Benton, and Andrew G Wilson. Residual pathway priors for soft equivariance constraints. In M. Ranzato, A. Beygelzimer, Y. Dauphin, P.S. Liang, and J. Wortman Vaughan, editors, Advances in Neural Information Processing Systems, volume 34, pages 30037–30049. Curran Associates, Inc., 2021.

---

### Decision · Program_Chairs · 2023-09-21

**Decision:**

Accept (poster)

**Comment:**

The paper is aimed at contributing to a growing body of work investigating the performance of ML models under various relaxed notions of symmetry. Central to the analysis is $Stab_\epsilon$, a notion relaxing the conventional form of group stabilizers allowing $\epsilon$-error. Classical results from statistical learning theory are used together with the latter to derive generalization and approximation bounds, and trade-off thereof.

The reviewers appreciated the problem the paper sets to address and found the bounds resulting by the use of $Stab_\epsilon$ enabling a more quantitative, if theoretical, exploration of the total error under relaxed symmetry, novel and intriguing. The authors are encouraged to incorporate the important feedback given by the knowledgeable reviewers (also make sure the correct NeurIPS format is used).